*Version of December 7, 2023*

# Radiative effect of thin cirrus clouds in the extratropical lowermost stratosphere and tropopause region

Reinhold Spang[1], Rolf Müller[1], and Alexandru Rap[2]

[1]Forschungszentrum Jülich GmbH, IEK-7, 52425 Jülich, Germany
[2]School of Earth and Environment, University of Leeds, Leeds LS2 9JT, United Kingdom

**Correspondence:** Reinhold Spang (r.spang@fz-juelich.de)

**Abstract.**

Cirrus clouds play an important role in the radiation budget of the Earth; nonetheless, the radiative effect of ultra thin cirrus clouds in the tropopause region and in the lowermost stratosphere remains poorly constrained. These clouds have a small vertical extent and optical depth, and are frequently neither observed even by sensitive sensors nor considered in climate
model simulations. In addition, their shortwave (cooling) and longwave (warming) radiative effects are often in approximate balance, and their net effect strongly depends on the shape and size of the cirrus particles. However, the CRyogenic Infrared Spectrometers and Telescopes for the Atmosphere instrument (CRISTA-2) allows ultra thin cirrus clouds to be detected. Here we use CRISTA-2 observations in summer 1997 in the northern hemisphere midlatitudes together with the Suite Of Community RAdiative Transfer codes based on Edwards and Slingo (SOCRATES) radiative transfer model to calculate the radiative effect
of observed ultra thin cirrus. Using sensitivity simulations with different ice effective particle size and shape, we provide an estimate for the uncertainty of the radiative effect of ultra thin cirrus in the extratropical lowermost stratosphere and tropopause region during summer and – by extrapolation of the summer results – for winter. Cloud top height and ice water content are based on CRISTA-2 measurements, while the cloud vertical thickness was predefined to be 0.5 or 2 km. Our results indicate that if the ice crystals of these thin cirrus clouds are assumed to be spherical, their net cloud radiative effect is generally positive
(warming). In contrast, assuming aggregates or a hexagonal shape, their net radiative effect is generally negative (cooling) during summer months and very likely positive (warming) during winter. The radiative effect is in the order of $\pm(0.1\text{-}0.01)$ W/m$^2$ for a realistic global cloud coverage of 10%, similar to the magnitude of the contrail cirrus radiative forcing (of $\sim 0.1$ W/m$^2$). The radiative effect is also dependent on the cloud vertical extent and consequently the optically thickness and effective radius of the particle size distribution (e.g. effective radius increase from 5 to 30 $\mu$m results in a factor $\sim 6$ smaller long and
shortwave effect respectively). The properties of ultrathin cirrus clouds in the lowermost stratosphere and tropopause region need to be better observed and ultra thin cirrus clouds need to be evaluated in climate model simulations.

## 1 Introduction

Cirrus clouds are an important contributor to the radiation budget of the Earth (e.g., Liou, 1986; Heymsfield et al., 2017). Despite recent progress in understanding cloud formation processes, aerosol-cloud interactions, and cirrus cloud radiative

effects (Forster et al., 2021), uncertainties for climate predictions are still large. From several positive feedbacks induced by doubling of $CO_2$, the cloud feedback has the largest spread between different GCMs and thus is the most uncertain (Vial et al., 2013; Boucher et al., 2013; Sherwood et al., 2020).

The classification of different cirrus cloud classes with respect to optical thickness ($\tau$) has been defined by Sassen and Cho (1992) and is based on lidar observations at visible wavelengths, where an optically thick cloud is defined for $\tau > 3$, opaque cirrostratus for $0.3 < \tau < 3$, transparent or thin cirrus for $0.03 < \tau < 0.3$, and subvisible cirrus (SVC) for $\tau < 0.03$. More recent observations with infra-red limb sounders found even optically thinner cirrus clouds in the range $10^{-6} < \tau < 10^{-2}$ in the tropopause region and partly significantly above the tropopause (Spang et al., 2002, 2008, 2015; Zou et al., 2020; Bartolome Garcia et al., 2021). These observations were obtained by space and airborne limb sounders. These clouds have been also detected by in situ particle measurements (Krämer et al., 2016) but are only observable to a moderate extent by space borne lidars (Davis et al., 2010; Spang et al., 2015; Balmes and Fu, 2018). To better discriminate the optically extremely thin clouds from most of the cirrus defined above, these clouds are referred to in the following as ultra thin cirrus (UTC), which is in line with the optical thickness definition ($\tau = 10^{-3}$ to $10^{-4}$) of the ultrathin tropical tropopause cirrus (UTTC) (Peter et al., 2003) detected with airborne lidar and in situ particle measurements. The dehydration potential of UTTCs in the tropics was shown by Luo et al. (2003). Cirrus clouds in the tropopause region may have a general and significant imprint on the water vapour amount in the stratosphere, and consequently via radiation effects of the stratospheric water vapour on the surface temperature (Riese et al., 2012).

Assuming UTCs at the tropopause (TP) to be a common cloud type, the question arises about the radiative effect of these clouds – cooling or warming. The imprint of UTCs on the radiative net effect of cirrus is not well constrained due to the difficulties to characterise microphysical as well as macrophysical quantities of UTCs. Hong et al. (2016) reported a detailed analysis of the ice cloud radiative effect over a wide range of optical thickness based on space borne Cloud–Aerosol Lidar and Infrared Pathfinder Satellite Observations (CALIPSO) lidar and CloudSAT radar data. They computed the cloud radiative effect (CRE) of ice clouds for a global multiyear climatology of retrieved ice water content, effective radius, and extinction covering low-level optically thick to high-level thin cirrus clouds. For the 2008 period, they found the warming effect (21.8 W/m$^2$) induced by ice clouds trapping longwave radiation exceeds their cooling effect (16.7 W/m$^2$) caused by shortwave reflection, resulting in a net warming effect (5.1 W/m$^2$) globally on the earth–atmosphere system. The study does not include the optically thinnest cirrus clouds like the lower-end in optically thickness of UTCs. These are hard to detect for CALIOP and may be underestimated in the dataset.

Davis et al. (2010) showed the difficulties to observe optically very thin cirrus clouds around the tropopause with the CALIOP lidar in comparison to in situ particle measurements. The analysis suggests that a majority (>50%) of SVCs around the tropopause ($\tau < 0.01$) could be unaccounted for in studies using CALIPSO data. Consequently, Hong et al. (2016) show only data for $\tau > 0.01$.

A similar study by Matus and L'Ecuyer (2017) is focusing not only on ice clouds but liquid, ice, multi-layer, and mixed-type clouds. Results of the global mean cloud radiative effects show only for ice a cooling effect and of comparable size to Hong et al. (2016) (3.4 W/m$^2$). Balmes and Fu (2018) show that the difficulties for the detection of very thin cirrus clouds still exist in

newer versions of the CALIOP cloud data products. They used ground based Raman lidar measurements and found significant deficiencies in the occurrence frequency compared to CALIOP (global estimates +0.13 - +0.17).

A more recent analysis of stratospheric ice clouds (SIC) based on the CALIOP cloud product shows rather high occurrence frequencies (2-20%) nearly all over the globe (Zou et al., 2020, 2022), with maxima in the tropics but local maxima at mid latitudes at the storm track regions, with a preference for the tropics but reasonable occurrence frequencies at mid and high latitudes (2-10%). In addition, the CALIOP shows less SICs at mid-latitudes than the Michelson Interferometer for Passive Atmospheric Sounding (MIPAS) on the Envisat satellite (IR-limb sounder) if both data sets are normalized in the tropics (Zou et al., 2020). Cirrus cloud occurrence frequencies retrieved with the CRyogenic Infrared Spectrometers and Telescopes for the Atmosphere (CRISTA) instrument showed for the first time on global scales significant numbers of cirrus occurrence at and above the tropopause (Spang et al., 2002, 2015) and present IR limb sounder as one of the most sensitive measurements for the detection of cirrus clouds. Due to its better vertical resolution than MIPAS these data allow a better and more sophisticated quantification of the radiative effect of the proven optically thinnest cirrus clouds.

So far it is unacknowledged if climate models would need to consider UTCs as a separate cirrus type. Some of the models show the capability to form optically and vertically thin clouds around the tropopause for a relatively coarse GCM resolutions (e.g. Gasparini et al., 2018), fine-resolution cloud models (Gasparini et al., 2022), and global storm resolving models (Nugent et al., 2022; Turbeville et al., 2022). The global radiative effect of these clouds is an open question, and the validation of cloud occurrence frequencies and cloud fraction compared with global measurements are still needed.

The present study is structured in the following way: Section 2 describes the instrument data and corresponding analyses and retrievals for setting up most realistic optically thin cloud profiles in the tropopause region at mid and high latitudes. Section 3 presents radiative transfer calculations with the SOCRATES model and the results in context of the warming and cooling potential of the corresponding macro and micro physical cirrus characteristics, followed by a discussion regarding limitations and uncertainties of the statistics as well as the estimation of the overall cloud radiation effect of the specific cirrus types.

## 2 Data sets and methodology

### 2.1 CRISTA instrument

The CRISTA instrument was flown twice on space shuttle missions in 1994 and 1997 in a free-flyer configuration (Offermann et al., 1999; Grossmann et al., 2002). During both missions the instrument made around 8 days of nearly global measurements in the mid-IR (5-15 micron), with still today an unprecedented horizontal resolution for a limb sounder. This was possible by the implementation of three telescopes with crossing viewing directions, which results in a rather dense measurement net (Fig. 1) - even in the tropics (not presented). With an along track sampling of 200/400 km and three viewing directions the typical gaps between the orbits in the equator and mid-latitude region are filled with measurements. Additionally, the pointing capability of the satellite allowed CRISTA-2 to extent the latitudinal coverage up to 74°N/S away from the more restricted coverage defined by the orbit inclination of 57° (fixed value for CRISTA-1 coverage). The rather good vertical resolution and

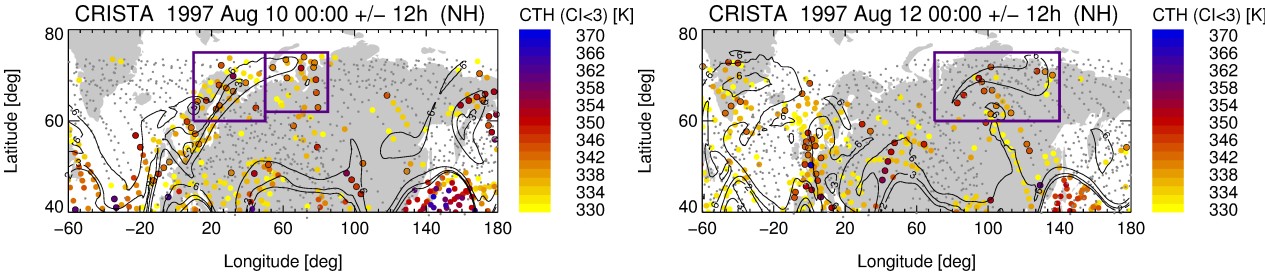

**Figure 1.** Regions of interest with high altitude cirrus clouds coverage for August 10 and August 12 1997 with high altitude cirrus clouds highlighted in the in the purple boxes. Cloud top heights (CTHs) are given in potential temperature coordinates by colored dotted symbols. Symbols with a black circle are highlighting CTHs above the lapse rate tropopause based on ERA5 reanalysis data. Grey dots represent non-cloudy observations. Black lines show contours of the potential vorticity (PV) at 350 K altitude for 2, 3, and 6 PV units, highlighting dynamical features like Rossby wave breaking events. The maps are adapted from Spang et al. (2015) where more details are presented.

sampling of 1.5 km and 2.0 km respectively, helps cirrus clouds to be detected and the exact location with respect to the tropopause to be determined with rather high accuracy (Spang et al., 2002, 2015).

The typical horizontal averaging along the line of sight of a limb sounder is in the order of 200-300 km, where the narrow
field of view helps to keep this uncertainty in an adequate range but this limits the accuracy of the retrieved limb ice water path (IWP) (see also Sec 2.3 for details) from CRISTA.

The exact position of the cloud along the limb path (line of sight) remains unknown in limb measurements and is not retrievable. Simplified assumptions, e.g. a fixed horizontal cloud extent, are necessary to solve this issue in a retrieval process for target parameter like IWC or extinction (e.g. Wu et al., 2008; Spang et al., 2015; Bartolome Garcia et al., 2021).
Modified instruments like limb imager allow substantial improvements of macroscopic cloud parameter with refined observation and retrieval techniques (e.g. 2D/3D and tomographic retrieval techniques, Ungermann et al., 2020). However, the CRISTA measurement capabilities are still unique for the limb IR technique. Extremely low cloud optical thicknesses are detectable which results in a very high detection sensitivity for IWC ($> 5 \cdot 10^{-6} \, g/m^3$), vertical IWP ($2 \cdot 10^{-4} \, g/m^2$), and extinction ($8 \cdot 10^{-4} - 10^{-2} \, \mathrm{km}^{-1}$) (Spang et al., 2012, 2015). The IWC value represents only 1/10 of the CALIOP detection
threshold (Avery et al., 2012) and the extinction threshold represents the upper limit of the measurement range of IR limb sounders. For extinctions $> 10^{-2} \, \mathrm{km}^{-1}$ the IR spectrum saturates, becomes optically thick in the limb, and the instrument loses sensitivity for optically thicker clouds.

## 2.2 SOCRATES radiation model

Radiative flux calculations have been performed using the offline version of the SOCRATES (Suite Of Community RAdiative
Transfer codes based on Edwards and Slingo) radiative transfer model (Edwards and Slingo, 1996) with six bands in the shortwave (SW), nine bands in the longwave (LW) and a delta-Eddington two-stream scattering solver at all wavelengths. This version has been used extensively in previous studies for calculating radiative effects from several atmospheric agents,

including contrails (e.g. Myhre et al., 2009a; Rap et al., 2010), water vapour (e.g. Myhre et al., 2009b; Riese et al., 2012; Kunz et al., 2013), ozone (e.g. Rap et al., 2015a; Riese et al., 2012), or aerosols (Rap et al., 2013, 2015b, 2018).

The model simulates ice clouds radiative effects using the Baran et al. (2014) parameterisation for ice clouds bulk optical properties. Radiative flux calculations are performed for specified ice cloud fraction, ice crystal effective radius and mass mixing ratio. Our sensitivity simulations consider three different ice crystal shapes: spherical particles, hexagonal cylinders (Yang et al., 2000; Rodríguez De León et al., 2018), and aggregates based on 83 representative size distributions measured during the CEPEX campaign Baran (2003). The diurnal variations in the SW are simulated by calculating the daily average SW radiative effects based on 24 instantaneous values (i.e. model runs with a 1-hour time resolution) using pre-calculated solar zenith angles (SZAs) at profile location. Clear-sky conditions below the cloud base are assumed in the radiative calculations. For the sake of a simplified setup we ignored multi-layer clouds. This disregards a potentially reduced radiative input in the longwave from underlying cold cloud tops with lower temperatures than the surface. Changes in the albedo with time or geographical location are considered by an incorporated time dependent 2D model of global albedo values.

## 2.3 Data preparation and model setup

In order to guide the model in a realistic clouds parameter space, the simulations were set up based on the CRISTA-2 measurements. IWC and extinction estimates, as well as accurate cloud top heights with respect to the tropopause height, are retrieved from the satellite and meteorological reanalysis datasets. ECMWF's fifth-generation reanalysis, ERA5 (Hersbach et al., 2020), is used for temperature, ozone and specific humidity information at the profile location of CRISTA-2. Additionally, ERA5 pressure and geopotential height are applied to transform between the vertical coordinate altitude for the satellite and pressure levels for the model. The derivation of lapse rate tropopause height and pressure from ERA5 is following the method of Hoffmann and Spang (2022).

The cloud detection capabilities and the high detection sensitivity for cirrus clouds by IR limb sounders, have been demonstrated in various studies (Spang et al., 2002; Massie et al., 2010; Spang et al., 2015; Zou et al., 2020) and for various instruments. The detection method is based on a color ratio of the emissions in a $CO_2$ and minor ozone band at a wavenumber region around 792 cm$^{-1}$ and an atmospheric window region at 830 cm$^{-1}$. The so-called cloud index (Spang et al., 2004) is high (CI>4) for optically thin and cloud free conditions and is shrinking to values close to one when getting optically thick in the limb direction. We applied a cloud index threshold of 3 for cloud top height detection, a robust threshold, applied and evaluated in various studies (e.g. Spang et al., 2004, 2012). The method was developed originally for the CRISTA satellite instrument but has been successfully adopted due to its high efficiency to other IR limb sounder, like the Michelson Interferometer for Passive Atmospheric Sounding (MIPAS) instrument (Fischer et al., 2008) on the Envisat satellite (Spang et al., 2004, 2012; Zou et al., 2020) or more recently on airborne instruments (Spang et al., 2008; Bartolome Garcia et al., 2021).

The CRISTA observations are used to define the macro and microphysical cloud parameter required for the SOCRATES run. In total 161 profiles with cirrus clouds around the tropopause have been selected from the CRISTA-2 dataset in the regions highlighted in Fig. 1. These regions are restricted to latitudes >60° N, which are not well covered by in situ instruments. The clouds have formed under specific dynamical situation where a Rossby wave breaking event over the Atlantic transported a

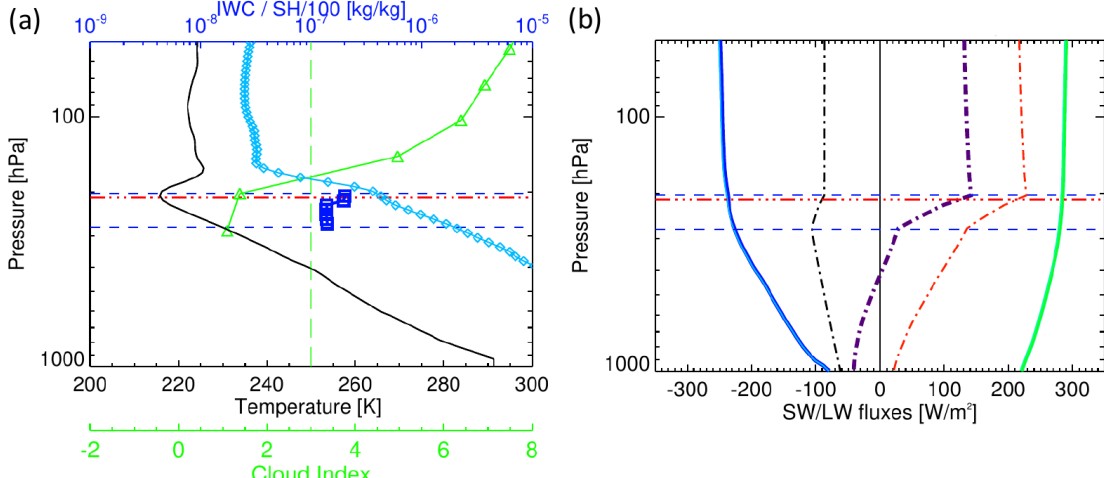

**Figure 2. (a):** Input profiles for the SOCRATES model runs for the 9th August 1997 13:47:13, 23.2°E and 68.6°N, with temperature (black) and SH (light blue) from ERA5, IWC (blue) retrieved for CRISTA-2 (like SH presented in kg/kg) and additional information like the cloud index profile for cloud detection (CI, in green - dimensionless - and threshold value CI=3 by a dashed green line) as well as cloud top and base height (blue dashed) and ERA5 based lapse rate tropopause (red dash-dotted) (following Hoffmann and Spang, 2022). All information are interpolated on the SOCRATES pressure grid. **(b):** Output profiles for spherical particles of SW (blue) and LW (green) all-sky and clear-sky fluxes by the model. The difference of all-sky and clear-sky (SW-black and LW-red) and the sum of SW and LW (purple) are shown by dash-dotted lines and are scaled by a factor of 100. Top of atmosphere total radiative effect results to $CRE_{SW}+CRE_{LW}$ = -0.86 + 2.14 = 1.28 W/m$^2$.

high water vapour amount to high latitudes over Europe and triggered the ice cloud formation in regions with sufficient cold temperatures, although the temperatures were usually to warm and additional temperature fluctuation by gravity waves would be necessary but were no resolved in the applied reanalysis data (Spang et al., 2015). To keep the amount of modelled profiles

for our study in a reasonable size as well as to choose well characterised UTC events we limited our analysis to two days and defined regions as highlighted in Figure 1. This restriction allows sensitivity studies over various micro and macro-physical parameters with the SOCRATES model.

Spang et al. (2012) showed, that the cloud index value in the optical thin part of cloud measurement correlates well with the limb integrated surface area density along the limb path (ADP) (equivalent to a limb IWP) and generated look-up tables

for various altitudes and latitudes. In addition, taken the effective radius of the particles into account (predefined for the model runs with 5, 10, and 30 $\mu$m) it is possible to estimate the mean IWC along the line of sight under the assumption of a tangent height layer homogeneously covered by the cirrus cloud. A constant path length through the cloud of 200 km has been applied. Usually, the path length depends on the tangent point altitude with respect to the real cloud top (not the detected instrument based cloud top height, because of the limited vertical resolution and sampling of the instrument the real CTH can be lower or

higher than the detected CTH). In addition, the cloud path length depends on the size of the field of view. Consequently, for a

horizontally and/or vertically large cloud extent the limb cloud path can become extremely long (150 to 400 km for 1 to 3 km respectively vertical resolution). In the current analysis, we are mainly interested in the cloud top region, up to 1-2 km below the cloud top, where the cloud is usually still optically thin and an effective cloud path of $\simeq$200 km is a reasonable assumption for the rather small field of view of 1.5 km of CRISTA (Spang et al., 2012, 2015). Finally, it needs to be highlighted that the cloud base height is difficult to retrieve for limb measurements. Obviously, for optically thick clouds in the limb (and nadir) direction the cloud bottom is not visible. In addition, for optically thinner clouds limited altitude coverage ($h_{min} >$ CBH) makes it impossible to determine a CBH for many limb sounders (e. g. MIPAS, CRISTA) especially for vertically thick clouds. Consequently, for the sensitivity study with SOCRATES we decided to use predefined cloud thicknesses ($\Delta z$) of 0.5 and 2 km.

Bartolome Garcia et al. (2021) showed for a newer and airborne based IR limb sounder the ability to retrieve CTH and CBH. The predefined thicknesses for the SOCRATES study are in the range of the maximum of the $\Delta z$ probability density ($\sim$ 600m) and $\Delta z$ =2 km is in the maximum range of the UTC thickness detected by the airborne instrument, where roughly 5-6% of the events show a $\Delta z > 2$ km.

The following procedure is applied for the interpolation of the CRISTA IWC to the SOCRATES grid. If more than one tangent height in a single profile is affected by clouds then the values (IWC) are used for the linear interpolation on the vertical grid down to cloud bottom altitude. If the predefined cloud thickness overlaps the minimum altitude of the measurement with IWC>0 then the model levels of the cloud are kept constant with the IWC of the lowest altitude.

In summary, IWC is estimated from the retrieved CTH and ADP or the ADP-equivalent limb IWP, where the effective radius and cloud thickness are predefined (0.5 and 2 km). The CRISTA measurements were taken in August 1997, which means under rather high solar zenith angle conditions with many hours of sunlight. For contrast we mirrored the CRISTA observation to February conditions to simulate contrapuntal winter-like events with respect to meteorological (ERA5) and solar conditions and used these for a rough estimate of the seasonal dependence of the CRE of UTCs (see Sec.4).

Particle shape and roughness are very important parameters for correctly modelling the radiative effect of cirrus clouds (Yi et al., 2013, and references therein). Cirrus particles can have very complex particles shapes (for a review see Lawson et al., 2019). Various studies show less complexity for the cold cloud top regions, especially in the tropical tropopause region where quasi spherical particles with radii smaller than 50 $\mu$m seems to dominate the particle size distribution and complex ice aggregates are less frequently observed (Woods et al., 2018). It is reasonable that cirrus at the tropopause and lower stratosphere (cloud top) at mid and high latitude have also less complex shapes than optically thick ice clouds in the free troposphere.

In this study we use properties of three particle shapes: (a) aggregates, a specific composition of ice crystal habits (Baran, 2003; Baran et al., 2014, 2016; Yang et al., 2005), (b) spherical ice particles as a simplification for the in situ observed quasi-spherical particles in the cloud top region, which are typically best described by droxtals (Yang et al., 2003; Zhang et al., 2004) or Chebyshev particles (Rother et al., 2006; McFarquhar et al., 2002), and (c) hexagonal cylinders (or columns, or prisms). Heymsfield and Platt (1984) reported that the ice crystals observed in high cirrus clouds (with cloud temperature <-50°C) were predominantly hallow or solid hexagonal columns. As described for example by Rodríguez De León et al. (2018) the parameterised optical properties for the hexagonal ice particle are based on Baran et al. (2001) and Yang et al. (2000) over a parameterised bimodal particle size distribution from McFarquhar and Heymsfield (1997).

Cloud optical depth ($\tau$) is an important parameter when investigating the cooling or warming potential of cirrus clouds (e.g. Hong et al., 2016). We computed $\tau$ from the CRISTA cirrus cloud detection by the CI-IWC relation prepared in look up tables of ADP or limb IWP versus CI with respect to cloud altitude and a further correlation between extinction $k_e$ and IWC/$R_{\mathrm{eff}}$ with $k_e = const \cdot \mathrm{IWC}/R_{\mathrm{eff}}$, with $const = 1.4 \cdot 10^3$, IWC in g/m$^3$, $R_{\mathrm{eff}}$ in $\mu$m, and $k_e$ in km$^{-1}$ (Spang et al., 2012). Finally, a simple integration of $k_e$ over the cloud layer thickness ($\Delta z$) results in the vertical optical depth $\tau = \int k_e dz$ which is used for the SOCRATES input (Figure 3).

The vertical information content of the ice parameters measured by CRISTA is limited. If more than one tangent height is affected by clouds in a single profile then the values (IWC and consequently $k_e$ and $\tau$) are used for the interpolation on the vertical grid of SOCRATES. If the predefined cloud thickness $\Delta z$ overlaps the minimum altitude of CRISTA then the model levels of the cloud are kept constant with the IWC of lowest altitude. The CRISTA based optical depth (Figure 3b) covers the full range of UTCs from extremely low $\tau < 0.0005$ up to a maximum of $\tau \simeq 0.05$, where the maximum is still in the range of SVCs.

For a crosscheck of the $\tau$ approach we also followed an estimation method for the optical depth by $\tau = 1.5 \cdot \mathrm{IWP}/(\rho_{\mathrm{ICE}} \cdot R_{\mathrm{eff}})$ (e.g. Baran and Francis, 2004) with $R_{\mathrm{eff}}$ in $\mu$m and $\rho_{\mathrm{ICE}}$ the mass density of ice in g/cm$^3$. PDFs of retrieved (Fig. 3a) and estimated (not shown) $\tau$ are looking very similar, which gives us confidence that the retrieved IWC, IWP and extinctions retrieved from CRISTA are in a reasonable range. Figure 3a and b show the distribution of IWP and optical thickness retrieved from the CRISTA data. The similarities in the PDFs highlight the close link between both parameters.

A summary of all different model scenarios (36 in total) for SOCRATES is presented in Tab. 1 together with the mean model results of the cloud radiative effect of all CRISTA profiles with cirrus clouds in the regions of interest (see Fig. 1) accounted for in the simulations. An example of the model input profiles (cloud index, temperature, specific humidity, and IWC) and the output profiles of SW and LW flux profiles for all-sky (as) cloudy and clear-sky (cs) none-cloudy conditions are presented in Fig. 2. CRE is defined as the sum of SW and LW effect of the all-sky minus clear-sky fluxes ($F$) at the top of atmosphere in W/m$^2$:

$$\mathrm{CRE} = (F_{\mathrm{SW_{as}}} - F_{\mathrm{SW_{cs}}}) + (F_{\mathrm{LW_{as}}} - F_{\mathrm{LW_{cs}}})$$

LW, SW and total flux (CRE) are saturating above the cloud top (<100 hPa) with virtually constant values (Fig. 2b), and a simple definition of the top of atmosphere (ToA) radiative effect is applicable by means of the pressure levels 20 to 0.6 hPa. The latter pressure value is the minimum pressure level in the model run.

Figure 3a gives an overview of the range and occurrence in the vertical IWP from the CRISTA estimates. The obviously very different vertical cloud thicknesses of 0.5 and 2.0 km result in a separation in the PDF of IWP with very small IWPs between $3 \cdot 10^{-3}$ and $5 \cdot 10^{-2}$ g/m$^2$ for the 0.5 km cloud layers and an IWP $\simeq 10^{-1}$ g/m$^2$ for the 2 km extended cirrus. All modelled respectively observed cirrus clouds are optically thin in the limb and nadir direction and will be invisible for most passive nadir instruments. The lowest IWP values are equivalent to an IWC of $10^{-5}$ g/m$^3$ which is one order of magnitude larger than the detection sensitivity of IR limb sounders (Spang et al., 2015) and close to the so far lowest IWC values observed in situ measurements ($10^{-6}$ g/m$^3$, Krämer et al., 2020).

**Table 1.** Median cloud radiative effect of 161 cirrus cloud profiles, as well as separated LW and SW effect in W/m$^2$ for various scenarios of the SOCRATES runs

| Scenario: | AUG | AUG | FEB | FEB | month |
|---|---|---|---|---|---|
| shape / $R_{\text{eff}}$ | 0.5 km | 2.0 km | 0.5 km | 2.0 km | cloud depth |
| hex / 5 $\mu m$ | *33* | *34* | *35* | *36* | *Scene ID* |
| | 0.56 | 3.86 | 0.24 | 1.71 | LW |
| | −0.78 | −6.33 | −0.16 | −1.09 | SW |
| | **−0.25** | **−2.18** | **0.04** | **0.59** | **CRE** |
| agg / 5 $\mu m$ | *25* | *26* | *27* | *28* | *Scene ID* |
| | 0.64 | 4.49 | 0.29 | 2.07 | LW |
| | −0.90 | −7.20 | −0.09 | −0.63 | SW |
| | **−0.29** | **−2.43** | **0.06** | **0.81** | **CRE** |
| sph / 5 $\mu m$ | *29* | *30* | *31* | *32* | *Scene ID* |
| | 0.57 | 3.99 | 0.29 | 2.18 | LW |
| | −0.20 | −1.71 | −0.04 | −0.29 | SW |
| | **0.35** | **2.28** | **0.26** | **1.86** | **CRE** |
| hex / 10 $\mu m$ | *17* | *18* | *19* | *20* | *Scene ID* |
| | 0.36 | 2.50 | 0.18 | 1.34 | LW |
| | −0.38 | −3.15 | −0.08 | −0.57 | SW |
| | **−0.05** | **−0.48** | **0.08** | **0.75** | **CRE** |
| agg / 10 $\mu m$ | *1* | *2* | *3* | *4* | *Scene ID* |
| | 0.40 | 2.72 | 0.19 | 1.46 | LW |
| | −0.44 | −3.44 | −0.09 | −0.63 | SW |
| | **−0.06** | **−0.64** | **0.09** | **0.81** | **CRE** |
| sph / 10 $\mu m$ | *9* | *10* | *11* | *12* | *Scene ID* |
| | 0.28 | 1.99 | 0.14 | 1.09 | LW |
| | −0.10 | −0.85 | −0.02 | −0.15 | SW |
| | **0.17** | **1.15** | **0.13** | **0.93** | **CRE** |
| hex / 30 $\mu m$ | *21* | *22* | *23* | *24* | *Scene ID* |
| | 0.13 | 0.92 | 0.07 | 0.51 | LW |
| | −0.11 | −0.93 | −0.02 | −0.17 | SW |
| | **0.00** | **0.01** | **0.04** | **0.33** | **CRE** |
| agg / 30 $\mu m$ | *5* | *6* | *7* | *8* | *Scene ID* |
| | 0.14 | 1.01 | 0.07 | 0.54 | LW |
| | −0.14 | −1.17 | −0.03 | −0.20 | SW |
| | **-0.01** | **-0.10** | **0.04** | **0.33** | **CRE** |
| sph / 30 $\mu m$ | *13* | *14* | *15* | *16* | *Scene ID* |
| | 0.09 | 0.65 | 0.05 | 0.35 | LW |
| | −0.03 | −0.27 | −0.01 | −0.04 | SW |
| | **0.06** | **0.37** | **0.04** | **0.31** | **CRE** |

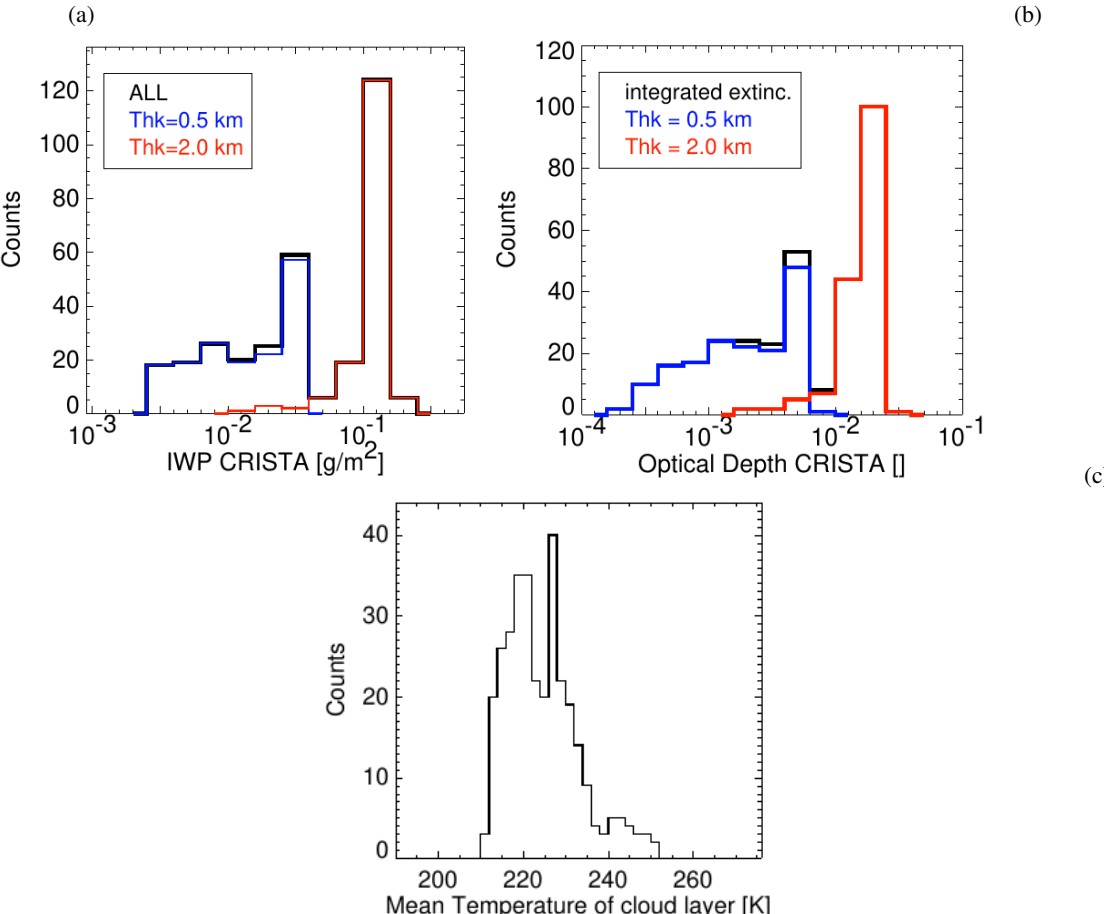

**Figure 3.** CRISTA based ice water path (a) and optical depth (b) distribution applied to the SOCRATES runs for $\Delta z$=0.5 km (blue) and $\Delta z$=2.0 km (red) and all together in black. (c) PDF of the mean cloud temperature used in the study for August conditions (ERA5 1997 data).

The mean temperature of the modelled cloud layers are for August in the range of 210 to 250 K (Fig. 3c) peaking around 220 K, which is a typical tropopause temperature at this time of season (s. Fig. 2, Hoffmann and Spang, 2022). We applied no pre-selection based on temperature for the CRISTA cloud detection, consequently a few (11 out of 162) lower altitude and warm cirrus are part of the SOCRATES calculations and are visible in the presented PDF at T >240 K. These events have been detected 2 to 4 km below the tropopause but at cloud top heights of 8 to 9 km, which is relatively close to the typical

tropopause height at the relatively high latitudes (60-70°N) of the measurements.

February temperature profiles have been selected from ERA5 and are not observational based (longitude and latitude of the August cloud observations). The corresponding estimated cloud temperatures are lower than in August (the maximum of the probability distribution peaks at 210 K, not shown). Although the selected temperature profiles for February may not

be realistic for the microphysical formation of a cirrus (e.g. frequent CTHs are modelled in February significantly above the
tropopause) it seams reasonable to use the profiles for T, SH, and $O_3$ to define the background atmosphere conditions for the
February scenarios in SOCRATES. However, these February simulations are only intended to provide rough CRE estimates
for the discussion on the overall differences between summer and winter conditions.

## 3 Cloud radiative effect of optically thin cirrus

An overview of CRE of the individual profiles is presented for selected model setups in Figure 4 by probability density functions
(PDF) for various $\Delta z$ and particle shapes (color-coded) and effective radius as well as for August (a, b) and February (c)
background conditions. Relatively broad distributions of negative CRE for hexagonal particles from -2 to 0.5 W/m$^2$ were found
for $\Delta z$ =2 km, changing to positive CRE for spherical particles (+0.5 to 2 W/m$^2$, Fig. 4a) for effective radius of 10 $\mu$m . Overall,
our simulations indicate contrasting behaviours between spherical and hexagonal particle shape and between August and
February conditions. Very similar distribution were found between hexagonal shapes and aggregates like shown for the large
mode ($R_{\text{eff}}$=30 $\mu$m ) but with similar agreement for both other size modes (Fig. 4b). Finally, for the small mode ($R_{\text{eff}}$=5 $\mu$m )
and February conditions the PDFs of spherical and hexagonal particles are shifted to positive CREs, only a minor contribution
of the scenarios with hexagonal particles is able to produce a net cooling effect. The results are discussed in more detail in the
discussion of Section 4.

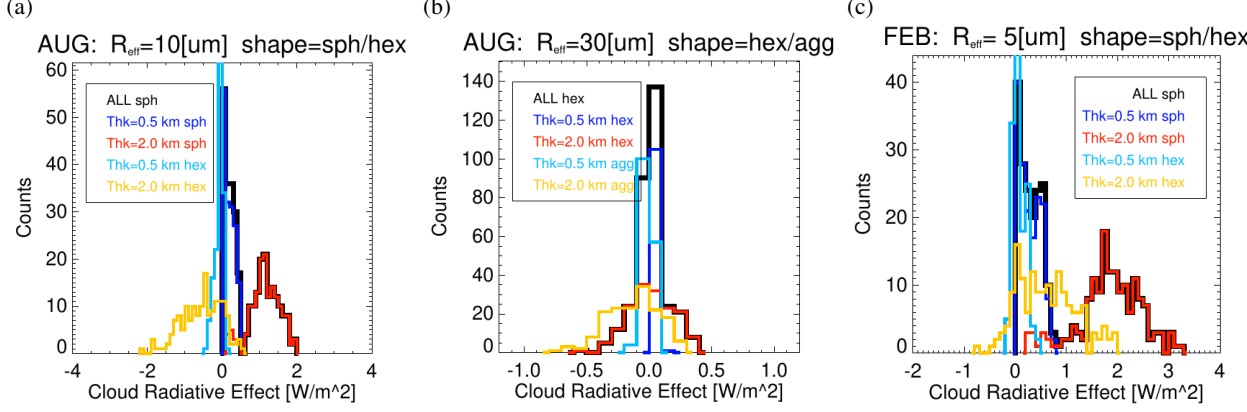

**Figure 4.** PDFs of CRE for various setups of the SOCRATES runs for 10 $\mu m$ effective radius of the particles for spherical and hexagonal
particle shapes (see color code) and (b) with effective radius of 30 $\mu m$ for hexagonal and spherical particles, both for August conditions and
(c) February background atmosphere for spherical and hexagonal particles. Separation in cloud thickness $\Delta z$ =0.5 km and $\Delta z$ =2 km in blue
and red for spherical/hexagonal/spherical (a-c) (the total of $\Delta z$ =0.5 and 2.0 km is highlighted in black and labeled with ALL) in comparison
to hexagonal/aggregates/hexagonal particles (0.5/2km in light blue/orange).

Figure 5 gives a better representation of the SW, LW and net effect with respect to the optical depth (Sec. 2.3) of the modelled
clouds. The left column shows CRE versus optical depth for 4 respectively 8 model scenes, always for $\Delta z$ = 0.5 and 2 km,

and August conditions with $R_{\mathrm{eff}} = 10\,\mu$m and shape aggregate (a, b), spheres (c, d), and hexagonal particle shape (e, f), and finally for the large particle mode 30 $\mu$m and hexagonal particle shape. The exponential growth of SW and LW effects with increasing $\log(\tau)$ is obvious for all setups, especially well pronounced for combination of $R_{\mathrm{eff}}$=10 $\mu$m and aggregates as well as hexagonal particles. SW and LW effect are nearly in balance and changes in shape or radius can significantly change the net
effect (black symbols in Fig. 5 a, c, e, and g) from cooling to warming and vice versa. The interplay of SW and LW effect is visualized in more detail in the right column of Figure 5 (b, d, f, and h). Deviations from the one-to-one line are highlighting the warming (above) and cooling (below) of the net cirrus CRE for each single cloud event modelled with SOCRATES. Smaller $\Delta z$ induces a very similar ratio between LW and SW effect, illustrated by rather similar gradients and only smaller amplitudes in the LW-SW correlation. For summer conditions and particle shapes of aggregates the short wave scattering effect can reach
a very large cooling (extended day length), especially for larger optical depth values ($\tau > 0.01$). This results in 10 scenarios (Tab. 1: Scene 01, 02, 05, 06, 17, 18, 25, 26, 33, 34) with an overall mean cooling potential (aggregates for 5, 10, and 30 $\mu$m, hexagonal shape only for 5 and 10 $\mu$m ) presented in Figure 5(a-d). However, even smaller effective radii ($< 10\,\mu$m) will result in even larger cooling effects for aggregates or similar complex particle shapes. In situ measurements show for optically thin cirrus clouds (SVC and/or UTC) typically $R_{\mathrm{eff}} <\simeq 100\,\mu$m and quasi-spherical with some plates and rare triagonal shapes
(Lawson et al., 2019). Thus, aggregates are not a very plausible particle shape for the optically thinnest cirrus clouds observed by CRISTA.

For spheres Figure 5 (c, d) compared to aggregates (a, b) or hexagonal particles (e, f) the SW scattering efficiency is less pronounced and the cooling effect is generally smaller in amplitude and usually smaller than the LW warming for spheres. Hexagonal shaped particles are more realistic for high altitude cirrus cloud particles (Heymsfield and Platt, 1984) although
usually observed at colder temperatures and not necessarily generally representative for UTCs at mid and high latitudes. Figure 5(e) represents an example the optical depth versus CRE and LW versus SW effect for 10 $\mu$m . The results are very similar to the aggregates, although the amplitude for SW, LW and total CRE are slightly larger for aggregates than for hexagonal particles. As a consequence, for hexagonal ice crystal shapes a few more profiles with LW > SW are present (f, h). The larger effective radius ($R_{\mathrm{eff}}$=30 $\mu$m (h, f)) results in significantly smaller surface area densities which reduces the SW effect. The LW
can now dominate the SW more frequently and results for nearly 50% of the profiles a net warming effect (h) compared to the 10 $\mu$m runs (f) with cooling for most of the profiles.

In addition, we present in Figure 6 the CRE versus optical depth relation for $R_{\mathrm{eff}}$= 5 $\mu$m and 10 $\mu$m of hexagonal and spherical particles for August and February conditions. The small mode shows again opposing effects for hexagonal and spherical particles, here with an extremely high SW cooling (up to 10 W/m$^2$) for hexagonal particles. Latter results for all
profiles in a net cooling and in contradiction for the spheres where the majority creates a net warming effect.

There is a strong $R_{\mathrm{eff}}$ dependency on the SW effect, where larger particles correspond to smaller surface area densities for constant IWC and consequently smaller SW scattering. Generally, 0.5 and 2 km cloud layers behave very similar to changes in the parameter settings (LW: orange and red, SW: lilac and blue in Fig. 5, Fig. 6). Whereby, $\tau$ is a significant factor of $\sim 4$ smaller in CRE for $\Delta z$=0.5 than for the $\Delta z$=2 km cloud layers.

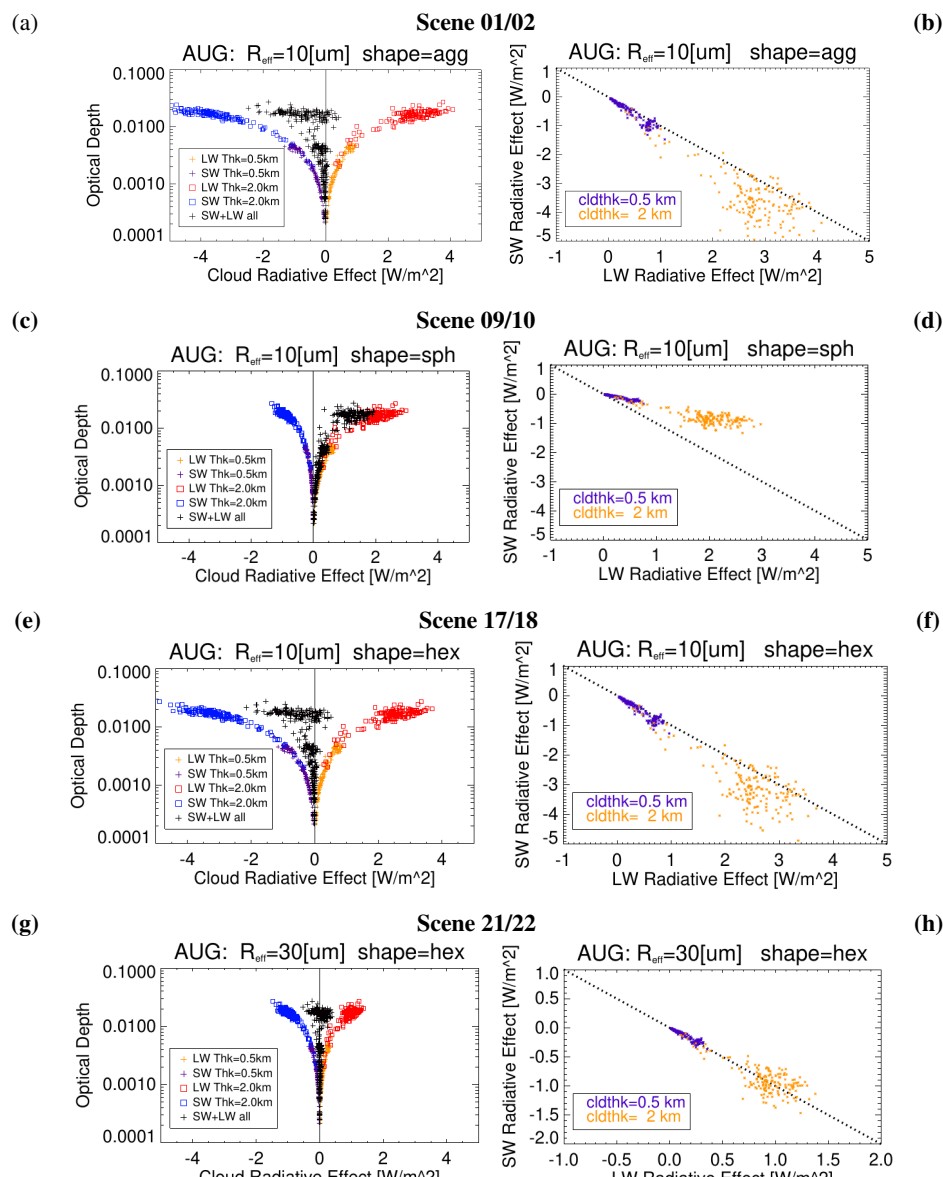

**Figure 5.** Cloud radiative effect for cirrus (colour coded LW, SW in blue/purple, red/orange (Δz = 2/0.5 km) and total in black) with respect to optical depth (left column) and LW versus SW radiative effect (right column). (a, b) aggregates, (c, d) spherical, and (e, f) hexagonal particles shape with $R_{\text{eff}} = 10\,\mu$m for August conditions, as well as (g, h) hexagonal particles with $R_{\text{eff}} = 30\,\mu$m. The black vertical line in the figures of the left column is highlighting CRE = 0 W/m$^2$. Attention, the y-axis range has changed for (h).

In addition, we tried to find a relation between CRE and macro-physical parameters like the distance to the tropopause as well the temperature difference between cloud top and surface temperature. Figure 7 presents only results for $R_{\text{eff}} = 10\,\mu$m (5

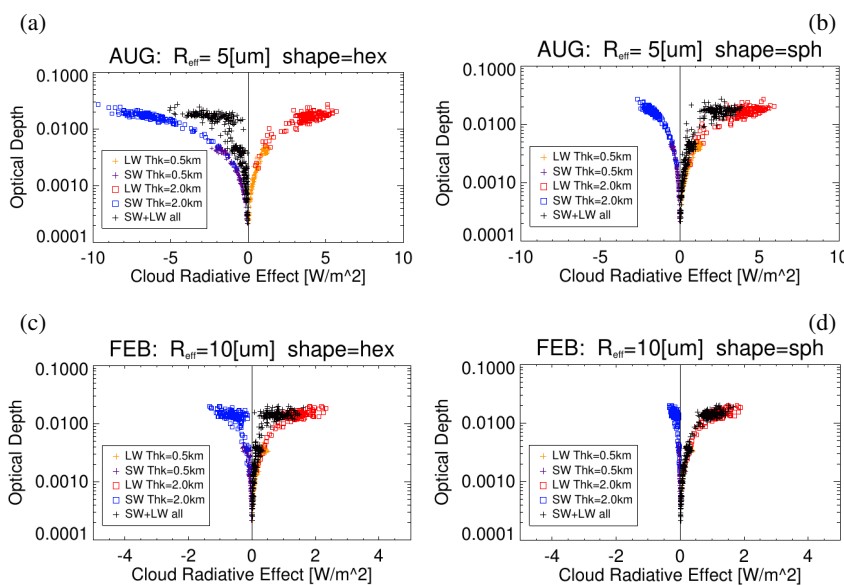

**Figure 6.** Cloud radiative effect for cirrus (colour coded LW, SW and total in blue/purple, red/orange ($\Delta z = 2/0.5$ km) and total in black) with respect to optical depth in August with $R_{\text{eff}} = 5\,\mu$m for hexagonal (a) and spherical particles (b). (c) and (d) show also hexagonal and spherical particles for $R_{\text{eff}} = 10\,\mu$m but February conditions (for details see text). The black vertical line in the figures is highlighting CRE = $0\,\text{W/m}^2$. Keep attention to the factor 2 expanded CRE range in (a) and (b).

and 30 $\mu$m differ only by a enhancement and reduction in magnitude respectively and no change in sign) for spherical (a) and hexagonal (b) particles for cloud top pressure minus tropopause pressure in pressure altitudes.

Although the interquartile range of the median values of CRE are rather large (error bars), for $\Delta z = 2$ km there is a distinctive correlation starting a few kilometers (2-3 km) below the TP up to 0.5 km above the tropopause. In addition, for spherical as well for hexagonal particle particle shapes the median indicates a weak negative gradient with cooling in CRE with larger distances above the tropopause (>0.5 to 2km). $\Delta z = 0.5$ km events show a negative gradient but with smaller tendencies in the range of $-1$ km to 3 km with respect to the tropopause. The gradient looks for aggregates and hexagonal particle shape opposite to spheres. Below the tropopause ($\simeq -1.5$ km) the gradient is negative for most of the scenarios.

Mean cloud temperature should have an imprint on CRE, because the emitted radiation of the cloud is directly related to the temperature. One of the major radiation effects by high cold clouds is a warming potential due to the emission at lower temperatures around the tropopause (e. g. Heymsfield et al., 2017). Cloud radiative effect with respect to the cloud top temperature (CTT) minus surface temperature at profile location are presented in Figure 8. CTT is not directly related to the LW effect of clouds, it is the temperature difference between the cloud top and the LW emitting layer in the atmosphere, 305    an underlying cloud layer or the surface temperature under clear-sky conditions (e.g. Gettelman and Sherwood, 2016). In addition, the optical thickness of the cloud layer is also an important parameter in this representation, the smaller the optical thickness the smaller the LW effect and consequently with smaller and less significant sensitivity to the temperature difference.

A completely transparent cloud ($\tau = 0$) has an negligible LW effect. This appears nicely in the different behaviour of the $\Delta z = 2$ km and dz=0.5 km scenarios in Figure 8, where the vertically thin clouds (red, scaled by a factor 4) show no or only weak trends in CRE with respect to the temperature difference. For all particle sizes and shapes for the vertically thick and consequently optically thicker cloud layers (blue) a linear relation becomes obvious, the smaller the negative temperature difference the smaller CRE. Although, the scatter - the error bars represent the interquartile range - is sometimes even large for the 2 km cloud cases, this appears only for temperature bins with small event numbers (N<5). In contrast, the $\Delta z = 0.5$ km scenes show always a relatively large interquartile range with respect to the median and only weak indication for similar trends (significantly smaller gradients) like in the $\Delta z = 2$ km scenes (Fig. 8a and d).

CRE versus optical depth similar to Figure 5 is presented Figure 9 with an additional binning along the corresponding y-axis and the computation of median and percentiles per bin for CRE. Median and the interquartile range are presented, with interquartile range as an error measure to better visualize the variability for constant optical depth. Median and interquartile range are better suited than mean and standard deviation to characterize data with high internal variability, extreme outliers, or skewed distributions. The median follows for spherical particles the exponential increase in CRE from the 0.5 and relatively small $\tau$ over the 2 km scenarios up to $\tau \simeq 0.02$ and CRE up to 2 W/m$^2$, whereby the aggregates show the nearly opposite behaviour in CRE, with smaller amplitude and larger scatter (interquartile range error bars).

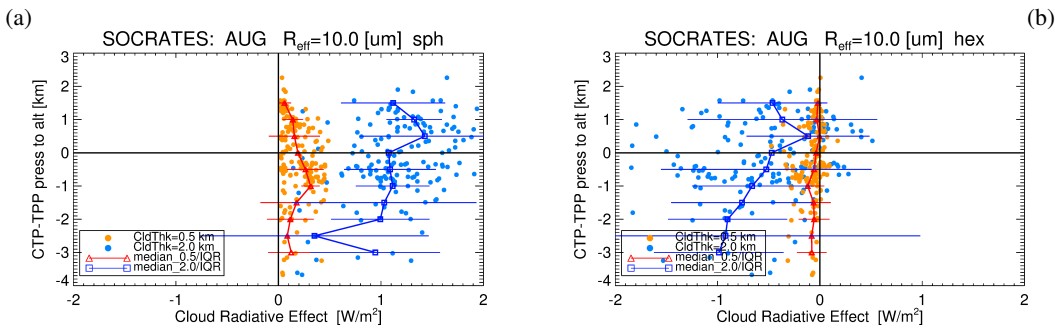

**Figure 7.** Cloud radiative effect with respect to the distance to the tropopause cloud top pressure minus tropopause pressure (CTP-TPP) for $R_{\text{eff}} = 10\,\mu$m spherical (a) and hexagonal (b) particle shape in the left and right column. Vertical line in middle highlights CRE=0 .

## 3.1 Statistics of modelled cloud radiative effect

Table 1 summarises all SOCRATES setups and mean results on LW, SW and net radiative effect. The median values of the selected areas (Sec. 2.3) give distinctive tendencies of the overall cloud warming or cooling effect for the UTCs. In total 161 cloud profiles have been selected from the CRISTA-2 data and represent variable conditions where optical thin cirrus clouds were formed at and above the tropopause north of 60°N. Using more realistic cloud occurrence frequencies would allow a more realistic assessment of the CRE of the UTCs.

The LW effect of scattering and trapping the IR radiation back into the direction of the ground is dominating the SW cooling effect for most of the model scenes. For spherical particle shapes all scenarios, Winter and Summer, result in a net warming

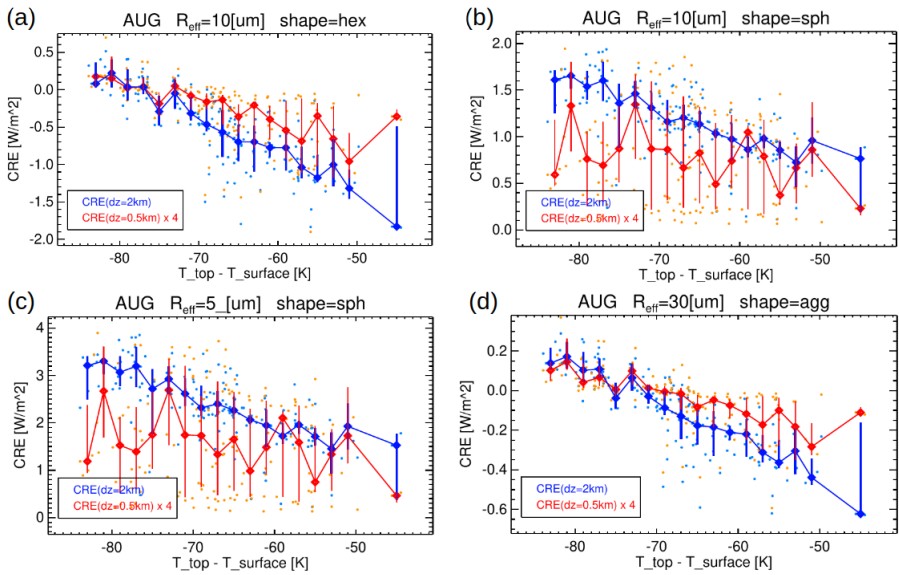

**Figure 8.** Cloud radiative effect with respect to cloud top temperature minus surface temperature (based on ERA5) for August, $R_{eff} = 10\,\mu m$ hexagonal (a) and spherical (b) particle shape, as well as for 5 $\mu m$ spheres (c) and 30 $\mu m$ aggregates (d). Cloud thickness is highlighted in blue for $\Delta z$ =2 km and red for $\Delta z$ =0.5 km. All numbers for $\Delta z$ =0.5 km events are scaled by a factor 4 in the CRE domain for simpler comparison. Median CREs for a temperature grid with a width of 2 K are superimposed by large symbols and error bars for the interquartile range.

effect in the range of 0.17 to 1.15 W/m$^2$ (AUG) and 0.13 to 0.93 W/m$^2$ (FEB) for $R_{eff}$ =10 $\mu m$. These are generally larger values than the mean effects for aggregates or hexagonal particles, especially for Summer conditions.

The LW warming is even more effective for larger particles where the resulting smaller surface area densities reduce the SW cooling. However, the SW cooling is also more effective for larger particles. Tab. 1 shows this fact for all simulations of
spherical particle, for example between 10 and 30 $\mu m$ effective radius a ~3 times higher SW and LW effect for the 10 $\mu m$ effective radius (e.g. factor 6 higher for 30 to 5 $\mu m$ .

The shortwave scattering effect depends strongly on the solar zenith angle and the duration of sunlit hours. For winter conditions the mean SW effect cannot compensate the LW effect for aggregates and spheres, whereby for summer conditions the aggregates can change the sign for CRE from warming to cooling. SW is enlarged from $-0.63$ to $-3.44$ W/m$^2$ and results
in a mean CRE of +0.81 (FEB) to $-0.64$ W/m$^2$ (AUG). This change of sign – from warming to cooling – does not appear for spherical particles. In contrast to aggregates the CRE stays positive from Winter to Summer and shows even slightly enhanced CRE (e. g. 0.93 to 1.15 W/m$^2$ for 10 $\mu m$ and $\Delta z$ =2 km). The SW effect of spherical particles is by far smaller than the one for aggregates.

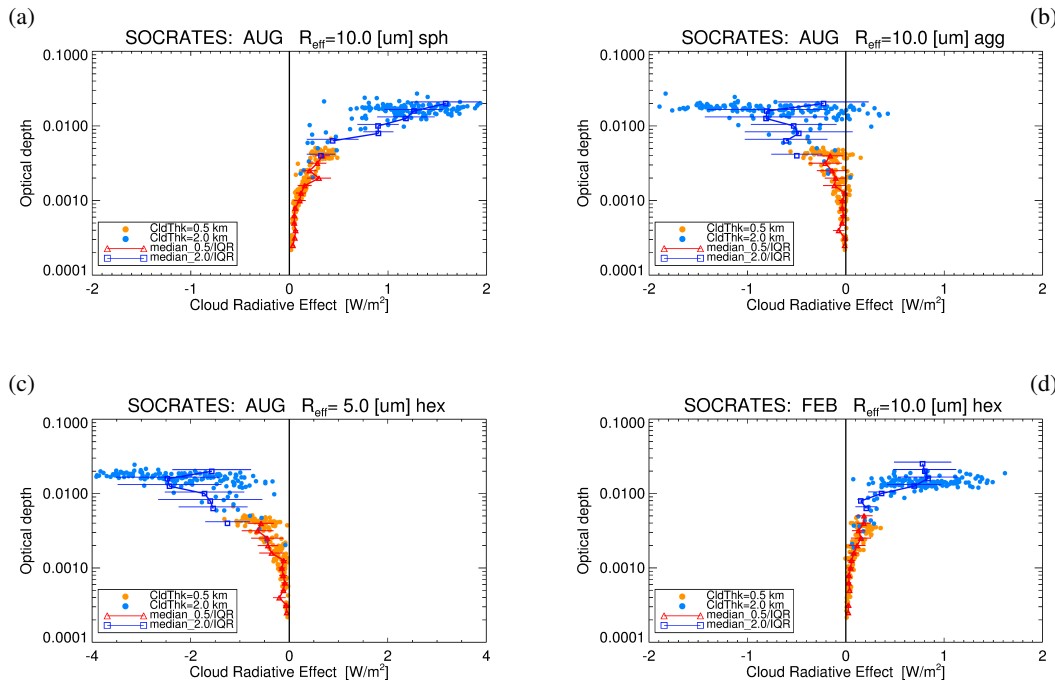

**Figure 9.** Cloud radiative effect with respect optical depth (a, b) for August, $R_{eff} = 10\,\mu$m spherical and aggregate particle shape in the left and right, and (c, d) with hexagonal particle shapes for $R_{eff} = 5\,\mu$m and August conditions as well as February for $R_{eff} = 10\,\mu$m. The error bars represent the interquartile range of the corresponding vertical grid box.

## 4 Discussion

### 4.1 Winter condition experiments

As mentioned in Section 2, we used the profiles for T, SH, and $O_3$ to define the background atmosphere conditions for February scenarios in SOCRATES. The CRISTA based input (CTH and IWC, indirectly IWP, extinction, optical depth) are simply mirrored into these February conditions. While these simulations do not represent actual February UTC CRE simulations, they are intended to provide a basis for discussions on the more general differences between summer and winter conditions.

For winter (February) conditions the short wave cooling becomes smaller due to reduced daylight hours (taking also into account the rather high geographical latitudes) which results in a general warming effect for all particle shapes and effective radii. The change from cooling to warming for aggregates and hexagonal particles with $R_{eff} = 10\,\mu$m is illustrated in Figure 6c and d. In addition, the albedo is varying for this change from a mean $\alpha = 0.15$ to $0.5$, which has a drastic effect on the LW radiance input coming from below the cloud. Similar to an underlying cloud, this can make a significant effect on the net SW/LW effect.

The change from net cooling to warming simulated for February conditions has been compared with long time record analysis of CALIPSO by Hong et al. (2016). It should be recapped that the cloud optical thicknesses <0.01 (mainly UTCs)

plays only a minor role for the total CALIPSO based CRE. Hong et al. (2016) showed the zonal mean seasonal variation of the total SW, LW and net effect of ice cloud radiative effect, latitudinal resolved (Fig. 4). For ∼50 to 75° N significant changes over the seasonal cycle, from cooling in summer (March to August) and warming in winter (September to February), are observable. A net warming effect of 10–20 W/m$^2$ is observed during winter and net cooling effect can exceed 30 W/m$^2$ in southern mid to high latitudes and 20 W/m$^2$ in the northern mid to high latitudes. In the tropics, strong net warming effect (10–20 W/m$^2$) persists over the whole year, mainly caused by high ice clouds with negligible seasonal variation in the zonal mean (Hong et al., 2016). Consequently, the CALIPSO results are in-line with the simplified mirroring of the CRISTA measurements from summer to winter conditions for UTCs. While, the amplitudes for UTCs are obviously much smaller, which is mainly a result of the different optical thicknesses of UTCs compared to ice clouds observed by CALIPSO. Tendency for a change from cooling to warming is identical.

## 4.2 Uncertainty in cloud thickness

Various settings of UTC clouds have been modelled with SOCRATES for a first quantification of cloud radiative effect of UTCs at mid and high latitudes. The original CRISTA-2 data have already been analysed regarding the cloud occurrence with respect to the tropopause by Spang et al. (2015). However, the CRE was not analysed for these unexpected observations of UTCs well above the tropopause and with unprecedented high occurrence rates at high northern latitudes. A comprehensive study was missing where the cooling or warming potential of these clouds is investigated.

For larger effective radii as well for vertical thinner cloud layers (0.5 km) we observe smaller cooling and warming effects in the mean SOCRATES results (Tab.: 1 and Sec. 3.1). The determination of $\Delta z$ would be an important parameter for a better future quantification of the CRE of UTCs. So far, there is no validated statistical information of the cloud thickness of UTCs available. Although the CALIOP data may miss a rather large part of the optically thinnest UTCs (Davis et al., 2010; Balmes and Fu, 2018), the dataset is still the best set available for such an analysis. Zou et al. (2020, 2021, 2022) have analysed cloud occurrence of stratospheric ice clouds (SICs) based on CALIOP data products. The Zou et al. studies have not focused on the vertical extent of these clouds. Zou et al. (2020) showed for mid latitudes that the IR limb measurements by MIPAS give higher occurrence rates on SICs than CALIOP, if both cloud occurrence frequencies are normalised in the tropics. This supports the notion that IR limb sounder are under specific conditions more sensitive in the detection of cirrus cloud than recent lidar in space instruments. This circumstance was already discussed in earlier studies (Spang et al., 2012, 2015; Bartolome Garcia et al., 2021).

The best information on the $\Delta z$ of UTCs retrieved from IR limb measurements is the more recent analysis of Bartolome Garcia et al. (2021) with data from the airborne GLORIA (Gimballed Limb Observer for Radiance Imaging of the Atmosphere) instrument (Riese et al., 2014) during the airborne campaign WISE (Wave-driven ISentropic Exchange) in the North Atlantic in September/October 2017. Due to the excellent vertical resolution and fine sampling using a 2D infrared imaging detector, the data achieve a vertical sampling and resolution of 140 m. For the first time cloud bottom information was retrieved from extinction profiles.

The analysis shows that in the region investigated by WISE there is a maximum likelihood in cloud extent $\Delta z$ of 500-625 m with likelihoods for $\Delta z > 1.5$ km of less than 25% of the peak likelihood and in total only 10% of all measured cloud events (Bartolome Garcia et al., 2021, Figure 8a). Note that for a typical $\Delta z$ of 0.5 to 1 km a lower CRE is calculated than for $\Delta z = 2$ km (Figure 5). Generally, optical depth is a good estimator for the strength of the CRE even at very low optical depth, but the cloud vertical extent is obviously a critical parameter for $\tau$ and CRE and there seams a functional relation between CRE, $\Delta z$, and $\tau$, respectively (Figure 5).

### 4.3 Estimates of UTC fraction and global CRE

Cloud top heights with respect to the tropopause show a week correlation with CRE in the SOCARTES results especially below the tropopause with a decrease in mean CRE with warmer cloud temperatures (Figure 7). The variability in the net effect is very large (profile to profile variability), which makes a it difficult to draw a final conclusion on a mean effect with cloud top position with respect to the tropopause, but we found clear indications for an enhanced warming effect for optically thicker events at higher altitudes with respect to the tropopause .

The CRE is not only affected by $\tau$ but the atmospheric background conditions (temperature profile, CTH location, surface albedo, microphysical quantities) can result in situations, where even similar background conditions and constant $\tau$ result in very different CRE. This makes the prediction of a net cooling or warming difficult. Even LW and SW effect turn out to be very different depending on the optical depth (Figure 5). Global measurements with a sophisticated occurrence frequency statistic, where the cloud detection is weighted with the covered area the by clouds, are necessary. The subsequent radiation model runs would then allow the overall CRE of UTCs in the tropopause region to be better quantified.

Hong et al. (2016) followed a very similar procedure with the combined CALIPSO/CloudSAT ice cloud data products (DARDAR: Delanoë and Hogan (2008) and 2C-ICE: Deng et al. (2010)). These data may not include the correct amount of UTCs due to a lack of detection sensitivity (see Sec. 1) and will consequently underestimate to some extent the total cloud coverage/occurrence of cirrus clouds. The Hong et al. results are presented for ice cloud optical depths $\tau > 10^{-2}$, a value where the limb measurements are starting to saturate. The Hong et al. analysis lacks the optically thinnest range of $\tau = 10^{-4} - 10^{-2}$ where IR limb sounders are most sensitive. A combination of IR limb and space lidar measurements would be an excellent combination to cover the full range of optical depth of cirrus clouds ($\tau = 10^{-4}$ to $> 20$) for a comprehensive view on the radiative effect of cirrus clouds. Hong et al. (2016) argue that subvisual cirrus ($\tau < 0.03$) display only weak SW and LW radiative effects because of their infrequent occurrence (global mean<5%, Hong and Liu, 2015) and this results in a global mean value of CRE = 0.05 W/m$^2$ (Hong et al., 2016, Table 4). However, as already argued above, the applied cloud dataset misses a significant number of UTCs and the overall radiative effect for SVCs and UTCs is underestimated. Our study shows that the radiative effect of these additional UTCs will be small but relevant, in the range of CRE median of -2.43 to 2.28 W/m$^2$ ($R_{\text{eff}}$=5 $\mu$m ) assuming 100% occurrence rate (with a single profile maximum of net cooling of -9 W/m$^2$ and net warming of $\sim$6 W/m$^2$, Fig. 6). For summer conditions only aggregates and hexagonal particles have the potential for a net cooling effect. If a more realistic UTC coverage of 10 (20)% is assumed (Zou et al., 2020) the CRE is reduced to a still not irrelevant range of -0.24 (-0.48) to 0.22 (0.45) W/m$^2$.

The numbers of the radiative forcing caused by UTC calculated here can be put into perspective of the radiative forcing determined for $CO_2$ and other greenhouse gases. Myhre et al. (2009b) analysed in a similar designed study SW, LW, and net radiative effect of a stratospheric water vapour (SWV) enhancement as well as effects of contrail cirrus in a comparison of various radiative transport models. An overall global change of SWV from 3.0 to 3.7 ppmv has resulted in a net RF (LW+SW) of 0.22 to 0.49 W/m$^2$ depending on the specific model. This is of similar value and range as for the 10 $\mu$m particle size in the CRE results for UTCs presented above (Tab. 1), with a mean 100% cloud-cover - clearly an overestimate for the real atmosphere. A typical contrail cirrus with optical depth of 0.3 and 100% coverage resulted in a CRE of $-14$ to $-7$ W/m$^2$ for high SZA (75°) and positive CRE of 12 - 23 W/m$^2$ for low SZA (30°). Similar to the UTCs focused on for this study, the CRE of optically thin contrails is very sensitive to the SZA, depending on the sunlight hours and the maximum SZA the overall CRE can change from negative to positive effects. Finally, Myhre et al. did a more realistic estimate of the global coverage of contrails and the resulting CRE. Four models deliver a rather good consistency in the range of 9.3 to 15 mW/m$^2$ (warming), which is significantly smaller than the estimates found above for 10 and 20% coverage of global UTCs with $R_{\text{eff}}$=10 $\mu$m . Only, if we further reduce the global coverage to 1% (to be consistent with the contrails coverage) the CRE of UTCs is in the range of -22 to 18 mW/m$^2$ for $R_{\text{eff}}$=5 $\mu$m and -5 to 9 mW/m$^2$ for 10 $\mu$m , where the warming of hexagonal shape and aggregates fits well with the contrails. However, these estimates indicate a significant larger CRE for UTCs than contrails.

Although the presented study specifies only first estimates for the CRE under various conditions, the results presented here show the priority to better constrain the amount (coverage) and vertical thickness of UTCs in the lower stratosphere and tropopause region. This would help to determine how important it is to consider UTCs in future climate model simulations. This would also help to prove if UTCs are at least to some extent considered in the current very common cloud dataset composed from radar and lidar space measurements (e.g. Delanoë and Hogan, 2008; Deng et al., 2010).

## 4.4 Possible impact on circulation patterns

Such optically thin clouds, like those reported above, may be important for the radiation budget of the upper troposphere and lower stratosphere and thus for modifying the mean tropopause temperature. They may thus indirectly control the amount of water vapour entering the stratosphere and influence circulation pattern. Radiative vertical flux profiles could be converted to heating rates (e.g. Kato et al., 2019), and a large cloud heating rate was shown to strongly modulate circulation patterns (e.g. Voigt et al., 2021; Gasparini et al., 2023). This activity could be the topic of follow-up work of the present study.

## 5 Summary and Conclusions

The present study reports the radiative effect of optically ultra thin cirrus clouds. Our sensitivity simulations with different ice particle sizes and shapes for those clouds provide an uncertainty range for their CRE during both summer and winter months. Cloud top height and ice water content are based on CRISTA-2 retrievals, while the cloud vertical thicknesses were assumed to be 0.5 or 2 km. Most of the model scenarios (Aug+Feb) result in a positive cloud radiative effect (warming) in the mean range of $-2.43$ to 2.28 W/m$^2$ for August conditions (Tab.1) and 100% cloud coverage. $R_{\text{eff}}$ is the leading parameter for

maximum cooling and warming, the smaller the particles the larger the amplitude of the CRE. Ice aggregates and hexagonal particles showed the ability to change the sign of CRE from cooling to warming when estimating February conditions with a simplified approach by mirroring the CRISTA observation into the winter atmosphere and considering albedo changes. Summer conditions with extended sunlit hours are producing large cooling rates over the day so that the SW cooling effect can dominate the LW warming effect, although, spherical particles always show a mean net warming effect.

Overall, the cooling and warming effects are nearly in balance for UTCs and the typical uncertainties of the various input parameters of the radiative transport calculation (e.g. particle type, effective radius or cloud layer thickness) makes it problematic to reliably quantify the net radiative effect. So far UTCs are an unnoticed cirrus cloud type in many fields from cloud microphysical modelling to the parameterisation of the formation processes for various types in global models. Further investigations on the particle shape, effective radius, cloud coverage, and vertical thickness of UTCs are suggested here to minimize the uncertainties in the radiative transport calculations. Finally, such work should allow a more accurate quantification of the cooling or warming potential of UTCs.

*Code and data availability.* SOCRATES is a model hosted by the UK Meteorological Office and accessible after registration from https://code.metoffice.gov.uk/trac/socrates/wiki. Model input (CRISTA-2 and meteorological reanalysis data) and output data by SOCRATES used in this study are available on request from the first author. In addition, all program codes in IDL used for the data analysis are available from the first author.

*Author contributions.* All authors designed the study. RS analysed and prepared the CRISTA data for the model runs. AR performed the SOCRATES runs and prepared the output files for radiative effect calculations. RS together with AR analysed the model output. All authors contributed to data interpretation and writing of the paper.

*Competing interests.* At least one of the (co-)authors is a member of the editorial board of Atmospheric Chemistry and Physics.

*Acknowledgements.* The authors like to thank D. Offermann and the former CRISTA team at the University of Wuppertal, Germany, for the realization of two successful instrument missions and the subsequent data processing. Thanks to the UK Meteorological Office for providing and supporting the SOCRATES radiative transport model. The authors are grateful to the European Centre for Medium-Range Weather Forecasts (ECMWF) for providing reanalysis data.

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
