# Peer review of "Radiative impact of thin cirrus clouds in the lowermost stratosphere and tropopause region"

_EGUsphere, 2023_

## Referee Comment (RC1)

Review of "Radiative impact of thin cirrus clouds in the lowermost stratosphere and tropopause region", by Spang, Müller and Rap.

This study examines the radiative properties of thin cirrus observed in the troposphere region by the CRyogenic Infrared Spectrometers and Telescopes for the Atmosphere (CRISTA-2) instrument during the second space shuttle mission in 1997. The article has similarities with the Spang et al. (2015) article, which I've gone through. What the article adds to that study is the use of the radiative transfer model SOCRATES, which is used to examine the bulk radiative flux properties of the thin cirrus layers, 161 cirrus, limb sampled on 9 August 1997. Basically, the question examined is whether the layers have a net positive or negative radiative effect.

Effective radii of 10 and 30 microns are assumed in the model, and ice particle aggregates-an 8-monomer hexagonal ice crystals, and spheres are assumed. With those assumptions, assumed cloud top height, and path length of 200 km for the CRISTA-2 measurements, the ice water content and ice water path are derived.

The article is clearly written, makes interesting use of the CRISTA measurements, and will be a valuable contribution. What I feel is needed is improved estimates of the effective radii and ice crystal shapes. Regarding the ice particle shapes, aggregates of ice crystals usually have sizes of 100-200 microns diameter and above. The assumed ice crystal aggregates are similar to bullet rosette ice crystals. And the spheres are designed to mimic droxtals. However, the ice particles in the TTL region are much more likely to be single crystals, such as hexagonal columns and trigonal crystals (see Heymsfield, 1986, JAS). Although in that study the temperatures were considerably below those studied here, the single crystal habit mode, and probably hexagonal plates or columns, is much more likely. Also see Bailey and Hallett (Q. J. R. Meteorol. Soc.(2002),128, pp. 1461–1483). An important reference is Kikuchi et al. https://doi.org/10.1029/2020JD033562. I also feel that the assumed sizes are too large. See the figure below for concentrations and size distributions in subvisual TTL cirrus (Heymsfield, 1986).

[Figure]

Number size distributions at six altitudes during rapid descent by WB-57 aircraft through subvisual cirrus near Kwajalein, Marshall Islands, on 18 December 1973 (Heymsfield and Jahnsen 1974). Thick lines denote ASSP measurements, thin lines 1DC measurements. Total concentrations in cm−3 appear in parentheses after height.

I recognize the desirability of using the SOCRATES model. Can any sensitivity studies be done, with smaller effective radii, for example. Can another radiative transfer model be used, that has the ability to do single columnar or hexagonal plate crystals incorporated? The changes will affect the net radiative effect of the cloud layers.

I have several minor comments. Just a few shown below:

Figure 2. What are the units of IWC.

122. "form" to "from"

127. What was the reason for choosing this path length?

185. seems

Andy Heymsfield, NCAR

---

## Author Comment (AC1)

**Radiative impact of thin cirrus clouds in the lowermost stratosphere and tropopause region**
**egusphere-2023-1234**

Reinhold Spang, Rolf Müller, Alexandru Rap

November 10, 2023

**1   General remark**

Thanks to both reviewers (Andrew Heymsfield and Blaz Gasparini) for their valuable comments and suggestions to improve the quality of the manuscript. Both reviewer had concerns regarding the particle types and effective radii used in the radiative effect calculations. We have enlarged the parameter space of our study by applying an even smaller effective radius of 5 $\mu$m on top of the 10 and 30 $\mu$m so far, and applied an additional cirrus particle type of hexagonal crystals. In addition, we highlighted more clearly that the analyses are not based on global observations and changed consequently the title of the manuscript by adding '... in the extratropical lowermost stratosphere and tropopause region'. Finally, we have formulated the abstract in a shorter and better readable form like suggested by reviewer 2. For details see the replies to the corresponding point-by-point comments of the reviewer as well as the new version of the manuscript.

We have highlighted the more detailed changes and new paragraphs in red in the revised version of the manuscript but also repeated most of the changes related to reviewer comments already here in this reply (roman font). Comments of the authors with respect to the reviewer comments are in italic letters.

**2   Point-by-point reply to the comments of reviewer 1: Andrew Heymsfield**

This study examines the radiative properties of thin cirrus observed in the troposphere region by the CRyogenic Infrared Spectrometers and Telescopes for the Atmosphere (CRISTA-2) instrument during the second space shuttle mission in 1997. The article has similarities with the Spang et al. (2015) article, which I've gone through. What the

article adds to that study is the use of the radiative transfer model SOCRATES, which is used to examine the bulk radiative flux properties of the thin cirrus layers, 161 cirrus, limb sampled on 9 August 1997. Basically, the question examined is whether the layers have a net positive or negative radiative effect.

Effective radii of 10 and 30 microns are assumed in the model, and ice particle aggregates-an 8-monomer hexagonal ice crystals, and spheres are assumed. With those assumptions, assumed cloud top height, and path length of 200 km for the CRISTA-2 measurements, the ice water content and ice water path are derived.

The article is clearly written, makes interesting use of the CRISTA measurements, and will be a valuable contribution. What I feel is needed is improved estimates of the effective radii and ice crystal shapes.

- Regarding the ice particle shapes, aggregates of ice crystals usually have sizes of 100-200 microns diameter and above. The assumed ice crystal aggregates are similar to bullet rosette ice crystals. And the spheres are designed to mimic droxtals. However, the ice particles in the TTL region are much more likely to be single crystals, such as hexagonal columns and trigonal crystals (see Heymsfield, 1986, JAS). Although in that study the temperatures were considerably below those studied here, the single crystal habit mode, and probably hexagonal plates or columns, is much more likely.

*We followed the helpful suggestions of the reviewer and added a third particle type for the model calculation (see following reply).*

- Also see Bailey and Hallel (Q. J. R. Meteorol. Soc.(2002),128, pp. 1461–1483). An important reference is Kikuchi et al. https://doi.org/10.1029/2020JD033562.

- I also feel that the assumed sizes are too large. See the figure below for concentrations and size distributions in subvisual TTL cirrus (Heymsfield,1986).

*We followed the suggestions by the reviewer and added a further effective radius of $5\mu m$ (now 5, 10 and 30 microns) and an additional particle shape of hexagonal ice particles (following Baran et al. optical parameterisation). The results are presented in the new version of the manuscript and show generally larger amplitudes for CRE, although aggregates and hexagonal ice particles show only slight differences in amplitude, with larger values for hexagonal particles.*

Page 6 we added following paragraph:

In this study we use properties of three particle shapes: (a) aggregates, a specific composition of ice crystal habits (Baran, 2003; Baran et al., 2014, 2016; Yang et al., 2005), (b) spherical ice particles as a simplification for the in situ observed quasi-spherical particles in the cloud top region, which are typically best described by droxtals (Yang et al., 2003; Zhang et al., 2004) or Chebyshev particles (Rother et al., 2006; McFarquhar et al.,

2002), and (c) hexagonal cylinders (or columns, or prisms). Heymsfield and Platt (1984) reported that the ice crystals observed in high cirrus clouds (with cloud temperature $< -50°C$) were predominantly hallow or solid hexagonal columns. As described for example by Rodríguez De León et al. (2018) the parameterised optical properties for the hexagonal ice particle are based on Baran et al. (2001) and Yang et al. (2000) over a parameterised bimodal particle size distribution from McFarquhar and Heymsfield (1997).

- I recognize the desirability of using the SOCRATES model. Can any sensitivity studies be done, with smaller effective radii, for example. Can another radiative transfer model be used, that has the ability to do single columnar or hexagonal plate crystals incorporated? The changes will affect the net radiative effect of the cloud layers.

*As described above we followed the suggestions by modifying the setup of the SOCRATES runs (in particular smaller radii) and have incorporated the results in the new manuscript.*

**2.1 Minor comments**

- Figure 2. What are the units of IWC.

*IWC is presented like SH in kg/kg, but SH is divided by 100. For better clarity this is now mentioned in the figure caption.*

- 122. "form" to "from"

corrected.

- 127. What was the reason for choosing this path length?

*The typical limb path length through the tangent height layer has been used in the analysis for the cloud path. The length depends on the vertical resolution of the instrument which defines the vertical coverage but indirectly by the observation geometry also the horizontal extent. This quantity is not retrievable from the measurement and need a priori information. We used the geometrical extent of the tangent height layer for a vertical field of view of 1.5 km (270 km) reduced it slightly to 200 km. In addition, we added a new paragraph giving more explanations on this issue in Section 2.3 of the manuscript:*

[revised manuscript text omitted]

---

## Author Comment (AC3)

**Radiative impact of thin cirrus clouds in the lowermost stratosphere and tropopause region**
**egusphere-2023-1234**

Reinhold Spang, Rolf Müller, Alexandru Rap

November 10, 2023

**1   General remark**

Thanks to both reviewers (Blaz Gasparini and Andrew Heymsfield) for their valuable comments and suggestions to improve the quality of the manuscript. Both reviewer had concerns regarding the particle types and effective radii used in the radiative effect calculations. We have enlarged the parameter space of our study by applying an even smaller effective radius of 5 $\mu$m on top of the 10 and 30 $\mu$m so far, and applied an additional cirrus particle type of hexagonal crystals. In addition, we highlighted more clearly that the analyses are not based on global observations and changed consequently the title of the manuscript by adding '... in the extratropical lowermost stratosphere and tropopause region'. Finally, we have formulated the abstract in a shorter and better readable form like suggested by reviewer 2.

For details see the replies to the corresponding point-by-point comments of the reviewer as well as the new version of the manuscript. We have highlighted the more extended changes and new paragraphs in red in the manuscript but also repeated most of the changes related to reviewer comments already here in this reply as well (blue). Comments of the authors with respect to the reviewer comments are highlighted in italic fonts.

**2   Point-by-point reply to the comments of reviewer 2: Blaž Gasparini**

- The manuscript by Spang et al. uses satellite retrievals of cirrus near the tropopause for a region in Eurasia to calculate its radiative effect. Given the uncertainties in particle radius and shape, they also perform several sensitivity experiments. This

is the first time that the top-of-atmosphere radiative effect of such optically thin, often unnoticed, cirrus clouds has been evaluated.

It is very valuable to determine the radiative effects of such thin clouds, which normally go unnoticed even by sensitive active sensors such as the CALIOP lidar. Not surprisingly, the study finds a very small net TOA effect compared to other cloud types. Their net effect is, however, strongly dependent on the assumption of particle size and shape.

- In particular, while the abstract reads as if the authors consider the radiative effect of such clouds worldwide, they ultimately perform the analysis on only a small subset of measurements in a region of northern Eurasia, for only one week. Climate variability can be extremely large on weekly timescales in a relatively small region of the extratropics, which is the main reason why I am not confident in the robustness of the generalization of the results in both space and time. Cirrus frequency may be different for other geographic locations and generally has a relatively strong seasonal cycle. Moreover, regions like the tropics are subject to substantially different meteorology, surface temperature, surface albedo... all of which need to be carefully considered (if they haven't already been).

  In addition, while using all available CRISTA measurements would only increase the sampling to 2 weeks, I don't see why not use all data from this otherwise relatively limited dataset.

*We agree on the concerns regarding generalization of the model results. We highlighted the geographical restriction in the abstract as well as in the title of the revised manuscript. However, it is not the intention of the study to model the global radiative effect of UTCs in detail. This fact is now well represented in our revised abstract. The study is designed as a sensitivity study to quantify the potential imprint of UTCs on the RE, typically not detectable by most satellite instruments and not specifically modelled in GCMs or CCMs. Various cloud parameters for model input are difficult to retrieve and quantify with remote sensing instruments. In addition, to retrieve cloud parameters for the complete measurement period of CRISTA (1 week) is not a simple extension of computer working time, hands-on work is necessary as well (IWC). However, we think such a dataset would be valuable for a future more global orientated study. The CRISTA mission sounds like a small dataset, but it isn't. Due to the moderate spectral resolution in the 4-15 micron range and three telescopes measuring in parallel, the information content was extremely high and unprecedented in the 1990s. Nearly 30 years later the amount of data and information is relatively easy to handle but still the number of limb profiles per day ($\sim$4000/day) is unprecedented by state of the art limb sounders.*

- Finally, I miss more discussion of why studying such clouds might be important at all. Given the much larger radiative importance of other cloud types, I don't think their TOA radiative effect is a strong enough motivation. However, even such thin clouds may be important in modulating the radiative budget within the

atmosphere and thus modifying the mean tropopause temperature. In this way, they may indirectly control the amount of water vapor entering the stratosphere (at least in the tropics) and influence circulation patterns. This may be their most important climatic role and should not be neglected in this analysis.

*We agree with the reviewer that the influence on the circulation and water vapour budget in general and in particular in the stratosphere are also good reasons to investigate into a better knowledge of UTC processes. Because of the gap of knowledge on formation processes and missing measurements of UTCs we designed a study focusing on UTC events detected in the LS, close to the tropopause, not detectable by most satellite based instruments, and at high latitudes where they are typically less frequently observed. In addition there was so far no study on these ultra thin cirrus clouds regarding the radiative effect and as a motivation for a first step, the study was designed to compute the cloud radiative effect of UTCs with reasonable accuracy. The study was not designed to investigate water vapour effect, which is also very important to investigate, but is focusing much more on the tropics, where the water vapour enters the stratosphere and any modification of the water phase has a strong imprint. However, the observed Rossby wave breaking event and enhanced water vapour in the vorticity streamer may also have an imprint on the lowermost stratosphere water budget (Spang et al., 2015).*

*In any event, in response to the comments by the reviewer, added in the introduction section more details, discussion and references on $H_2O$ feedback and circulation issues (e.g. Luo et al., 2003; Riese et al., 2012) - UTLS water vapour feedback on surface warming/cooling.*

**2.1   Additional general comments**

- 1. Ultra Thin Cirrus (UTC) are (most likely) standard representatives of the in situ formed cirrus, which happen to be too thin to be detected by most sensors. As such, there is no need to consider them as a separate cirrus type in climate models. While climate models in general struggle to correctly represent cirrus clouds, I see no evidence of climate models being particularly bad at simulating UTC. See, for example, examples of such thin clouds for coarse global climate models (e.g., Gasparini et al., 2018), fine-resolution cloud models (e.g., Gasparini et al., 2022, appendix), and global storm models (e.g., Turbeville et al., 2022, Nugent et al., 2022). In fact, models sometimes simulate a large number of such extremely thin cirrus. Moreover, given the inability of most satellite retrievals to detect UTC (and thus validate models), and given its small radiative impact, UTC can generally be neglected in standard climate model applications.

  Note: there may be other, better references compared to those listed

*Thanks to the reviewer for the detailed information on the status of the modelling community with respect to optically thin cirrus in the UTLS in cloud schemes of GCMs.*

*We think this is a valuable information and used it to improve the introduction section. Therefore, we have used the arguments to prepare a small section about the status of cloud models in GCMs for the introduction section:*

So far it is unacknowledged if climate models would need to consider UTCs as a separate cirrus type. Some of the models show the capability to form optically and vertically thin clouds around the tropopause for a relatively coarse GCM resolutions (e.g. Gasparini et al., 2018), fine-resolution cloud models (Gasparini et al., 2022) and global storm resolving models (Nugent et al., 2022; Turbeville et al., 2022). The global radiative effect of these clouds is an open question, and the validation of cloud occurrence frequencies and cloud fraction compared with global measurements are desired.

*We still believe it is important to consider observations of UTC (as they are available from CRISTA). If the effect of UTC is adequately quantified, as tried for the first time in this study, then a discussion about the need to consider or to ignore this cloud type is more meaningful. If the microphysical models or parameterisation for ice formation in climate models are able to produce optical thin cirrus clouds then it would be important to validate these clouds to ensure that the same physical process is responsible for and to show macrophysical quantities (e.g. cloud occurrence, spatial distribution etc.) either in or not in agreement with measurements.*

- 2. Is the main scientific motivation for studying UTCs really their (very small) net TOA radiative effect? At least for the tropical subset of UTC, they may be more important because of their role in stratospheric water transport (e.g., Luo et al., 2003).

*Yes and No! We now know the 'relatively' small effect of UTCs, but not before this study. So it is important to study the radiation effect of UTCs. The potential effect of UTCs on stratospheric water vapour is now highlighted in the introduction section:*

Luo et al. (2003) have shown the dehydration potential of UTTCs in the tropics. Consequently, cirrus clouds in the tropopause region may have a general and significant imprint on the water vapour amount and in the stratosphere, and consequently via radiation effects of the water vapour on the surface temperature (Riese et al., 2012).

- 3. What is the value of using aggregates to estimate radiative uncertainties, as it is almost certain that such high, thin cirrus are not composed of aggregates? Wouldn't it be of more value to consider the more common columns, rosettes, or budding rosettes instead (as per Lawson et al., 2019)?

*It is correct that aggregates are rather unlikely, although in a first sensitivity study it is important to cover a certain range of variability in microphysical cloud parameters, even in direction of extreme particle shapes. However, in addition to the shapes considered previously and for a better better representation of potential particle types in UTCs we added SOCRATES calculations for hexagonal ice crystals:*

In this study we use the scattering properties of three particle shapes: (a) aggregates, a specific composition of ice crystal habits (Baran, 2003; Baran et al., 2014, 2016; Yang et al., 2005), and in contrast (b) spherical ice particles as a simplification for the in situ observed quasi-spherical particles in the cloud top region, which are typically best described by droxtals (Yang et al., 2003; Zhang et al., 2004) or Chebyshev particles (Rother et al., 2006; McFarquhar et al., 2002), (c) hexagonal cylinders (or columns, or prisms). Heymsfield and Platt (1984) reported that the ice crystals observed in high cirrus clouds (with cloud temperature $< -50°$C) were predominantly hallow or solid hexagonal columns. As described for example by Rodríguez De León et al. (2018) the parameterised optical properties for the hexagonal ice particle are based on Baran et al. (2001) and Yang et al. (2000) over a parameterised bimodal particle size distribution from McFarquhar and Heymsfield (1997).

- 4. I am not convinced of the value of the mirrored February analysis. Clouds exhibit strong seasonality in mid and high latitudes. Therefore, it is hard to expect the same clouds to occur in winter. Also, in winter, the tropopause is lower, so clouds must also shift to lower altitudes.

*In principle, we agree with the reviewer on summer/winter differences. However, we think these winter-setups for the model have their justification for testing the sensitivity of the CRE of UTCs with respect to the amount of daylight, solar zenith angle variability, albedo changes and variability in the meteorological conditions.*

- 4.1 Moreover, were the possible differences in surface albedo between summer and winter considered? The selected regions are most likely snow-covered in February. This further diminishes their SW CRE.

*Albedo changes have been taken into account. You are right, at these high latitudes and snow cover in winter the albedo is changing extremely between for summer and winter conditions and strongly influences the SW CRE. This is described now in Section 2.2.:*

Changes in the albedo with time or geographical location are considered by an integrated time dependent 2D model of global albedo.

- I therefore think this should be removed from the result section. Such results may better fit in the discussion section, which may be a good place for such speculative results.

*We argue that the February – winter like – calculation are a meaningful part of the study, although, we applied simplifications, e.g. cloud top heights are not corrected with respect to the tropopause. Consequently, the conclusions are limited in the force of expression. However, we considered the changing sunlight hours and also the local and seasonal albedo changes based on a climatology implemented in the SOCRATES setup. We agree, that parts of this analysis are better placed in the discussion section and moved the corresponding paragraphs to section 4.*

- I would instead appreciate it if the authors showed the heating rates (units K day-1) in the result section. I am mentioning that particularly as it is very straightforward to transform radiative flux vertical profiles in W m-2 to heating rates. High cloud heating rate was shown to strongly modulate circulation patterns (e.g. Voigt et al., 2021, Gasparini et al., 2023 and references therein).

*While we agree that investigations on the heating rates and associated circulation changes is potentially very interesting, this is currently beyond the scope of our study. We have already substantially expanded our analysis in this revised version (by performing and analysing extra ice crystal shape and size simulations). Properly addressing the heating rate analysis would require substantial new work that could in itself constitute the main focus of a separate study.*

- 5. I didn't understand whether cloud geometrical thickness can be estimated directly from the CRISTA measurements. If so (as the selected cloud in Figure 2 suggests), why did the authors use predefined cloud thickness of 0.5 and 2 km in their analysis?

*Cloud geometrical thickness is a difficult parameter to retrieve with limb measurements especially with IR emission. The cloud top is easy to identify, in Fig. 2 where the cloud index show a strong gradient and in direction to small CI values (CI<3 to 1.5), although the detected cloud top height is influenced by the vertical field of view of the instrument (e.g. Spang et al., 2015). The greater the FOV the greater CTH uncertainty. A retrieval of cloud base heights depends on more quantities. Although the CI profile like in Fig. 2 suggests a layer like structure, there is more than one possibility to reproduce such a profile. It can be influenced by the optical thickness of sub-layers of the cloudy scene and the vertical thickness and horizontal extent of the cloud.*

*In addition, due to the limited vertical coverage of CRISTA (just a few kms below the tropopause on 2-3 tangent points, with 2 km vertical sampling) it is frequently not possible to estimate the lower boundary for cloud layer thicknesses of >2 km. To obviate this restriction of the CRISTA measurements we decided to use predefined geometrical thicknesses on both extreme sides, 0.5 and 2 km for optically thin to thicker cirrus conditions (but still optically thin in the IR limb and nadir direction) for the SOCRATES runs.*

**2.2 Specific comments**

- Page 1, abstract: Authors could improve the readability of the abstract by avoiding most abbreviations here and defining them later in the text.

*We agree on the suggestion of the reviewer and minimized the amount of acronyms, although this is not always applicable for example if a long satellite or instrument name is repeated a couple of times in the abstract.*

- Page 1, abstract: I strongly suggest that the abbreviation "RE" be changed to "CRE" for consistency with current atmospheric and climate science literature. I don't see an urgent need for the additional abbreviation "cirrus radiative effect".

*We followed the suggestion and are using now only the abbreviation CRE for cloud radiative effect, and deleted respectively replaced RE throughout the manuscript.*

- Page 2, line 28: Foster et al., 2021 may not be the best reference for that statement. If I don't miss something, it mentions the uncertain cirrus response to global warming only related to their very uncertain anvil area cloud feedback, which is not relevant for this study.

*We changed the wording of the statement and added two further references:*

Despite recent progress in understanding cloud formation processes, aerosol-cloud interactions, and cirrus cloud radiative effects (Forster et al., 2021), uncertainties for climate predictions are still large. From several positive feedbacks induced by doubling of $CO_2$, the cloud feedback has the largest spread between different GCMs and thus is the most uncertain (Vial et al., 2013; Boucher et al., 2013).

- Page 2, line 44: . . . they computed cirrus cloud radiative effect. . .
  I think the cited study considered all ice-containing clouds, not only high-altitude clouds; I therefore suggest replacing it with "ice cloud radiative effect" instead of "cirrus radiative effect".

*We corrected this statement accordingly.*

- Page 2, line 48:

  CALIOP can probably detect some of the thinnest cirrus considered in this study. There seems to be some overlap at cloud optical depths of 0.01. I would rewrite the sentence with a weaker statement.

*We changed the wording accordingly:*

The CALIOP study by Hong et al. (2016) does not include the optically thinnest cirrus clouds like the lower-end in optically thickness of UTCs. These are hard to detect for CALIOP and may be underestimated in the dataset.

- Page 2, lines 50-52: Beyond Davis et al„ 2010, Balmes and Fu, 2018 show a comparison of 2 ground-based Raman lidars with CALIPSO, showing the limit to CALIPSO detection

  In addition, Matus and L'Ecuyer, 2017 is another good CloudSat-CALIPSO reference for ice cloud radiative effect.

*Thanks for the excellent references. We incorporated the references in the introduction section (see revised version).*

- Page 3, section 2.1: A global map with CRISTA measurements may help in qualitatively understanding the spread of its retrievals

*We stay in the new version with the geographically restricted regions with UTCs and don't think that a global map highlights any more the spread/variability of the CRISTA retrieval. CRISTA delivers CTH as shown in Fig. 1 as well as IWC (IWP, optical depth) information (Fig. 2). Variability and spread of these parameters are presented as a function of various parameters in Fig. 3, Fig. 5 to Fig. 9.*

- Page 3, line 81: I don't understand the meaning of the sentence that begins with "Where along the line of..."

*We have described the limb technique and its distinctive geometrical features now in more detail.*

Where along the limb path (line of sight) the cloud is located, for example in front of or behind the tangent point, and how long the cloud is extended along the line sight is unknown in limb measurements and cannot be retrieved. Simplified assumptions, e.g. a fixed horizontal cloud extent, are necessary to solve this issue in a retrieval process for target parameter like IWC or extinction (e.g. Wu et al., 2008; Spang et al., 2015).

- Page 4, Fig. 1 caption: What are the contour lines representing? Is the region of interest the whole map section or only the three pink rectangles?

*We have now included more information in the Figure caption of the revised version:*

Regions of interest with high altitude cirrus clouds coverage for August 10 and August 12 1997 with high altitude cirrus clouds highlighted in the in the purple boxes. Cloud top heights (CTHs) are given in potential temperature coordinates by colored dotted symbols. Symbols with a black circle are highlighting CTHs above the lapse rate tropopause based on ERA5 reanalysis data. Grey dots represent non-cloudy observations. Black lines show contours of the potential vorticity (PV) at 350 K altitude for 2, 3, and 6 PV units, highlighting dynamical features like Rossby wave breaking events. The maps are adapted from Spang et al. (2015) where more details are presented.

- Page 4, Fig. 1: The detected thin cirrus seem to be occurring above mountains, likely connected to orographic mountain waves? Is this true? Are mountains a preferred location of very thin cirrus near/above the tropopause?

*This is an interesting topic, but not the focus of our study. There are indications for a link between mountain waves / gravity waves and high altitude cirrus clouds. Zou et al.*

*(2020) have detected stratospheric ice clouds (SIC) based on CALIOP data and analysed in a follow-up study (Zou et al., 2021) the link between SICs, deep convection events and gravity wave activity over North America based on AIRS data. Both processes can explain up to >70% each of the SIC observations depending on the season and defined sub-region (e.g. Great Plains, Eastern Canada, and Northwest Atlantic).*

*In the selected examples for CRISTA we cannot rule out a link to GWs, but the main trigger is very likely the Rossby wave breaking event over the North Atlantic with transport of relatively high water vapour to high northern latitudes. Although, trajectory studies showed that the temperatures are usually to high for ice formation (Spang et al., 2015). Unresolved GWs in the reanalysis data (ERA interim) over the Scandinavia or the Ural mountains may create the temperature variations needed to trigger ice formation.*

**Section 2.2:**

- Do you consider clear-sky conditions below the detected cirrus when calculating radiative fluxes? Please clearly state that and discuss the limitations of such an assumption!

*Yes, we assume clear sky conditions below the UTC, and we agree this is definitely an important information for the reader. We followed the suggestion and highlighted this at end of section 2.3 in detail.*

Clear-sky conditions below the cloud base are assumed in the radiative calculations. For the sake of a simplified setup we ignored multi-layer clouds. This disregards a potentially reduced radiative input in the LW form underlying cold cloud tops with lower temperatures than the surface. Changes in the albedo with time or geographical location are considered by an incorporated time dependent 2D model of global albedo values.

1. Is there a lower (altitude) bound to cirrus detection? Is there any lower bound considered to avoid clouds that occur at temperatures, where water can also co-exist with ice (e.g. T>-38℃) ?

*We didn't include a lower altitude bound for the detection. The CRISTA data has its own lower limit as part of the vertical sampling, where lowest tangent height of a profile has a minimum altitude of 6-8 km at high latitudes and 12-14 km in the tropics. This fact is described in the cloud thickness discussion in Section 2.*

2. How are thicker clouds treated? Are only their tops considered, or are they fully discarded from the data?

*Vertical thicker clouds than 2 km are not considered in SOCRATES runs at all, as well as underlying clouds. Only 0.5 km and 2 km thickness of the UTCs have been modelled with SOCRATES. Section 2.1-2.3 are presenting now the details. The observations may include vertical thicker clouds but due to the limited vertical sampling with respect to*

*the tropopause (only a few ~2 to 6 km) the cloud base of thicker clouds is usually not sampled.*

- Page 5, Fig. 2: The chosen cirrus seems pretty thick in terms of cloud geometry, given that the UTC are expected to be very thin.

  Also, for better visibility, I suggest cutting the panel (a) at 200 K, and the panel (b) at +/- 300 (or 350) W m-2.

*We followed the suggestions and changed the figure axis accordingly. The presented cloud is definitely not optically thick in the nadir direction. The optical thickness is retrieved from CRISTA measurements and is not predefined. The vertical cloud thickness ($\Delta z$) is for this example $\Delta z$=2 km which is the upper defined limit. The limited capability to retrieve cloud bottom information is now described in more depth in Section 2.3.*

- Page 6, line 154: Are the cloud properties assumed to be constant in vertical? Please mention that explicitly and explain the rationale for it.

*There is only a limited vertical information in the ice parameter of CRISTA. If more than one tangent height is affected by clouds in a single profile then the values (IWC) are used for the interpolation on the vertical grid of SOCRATES. If the predefined cloud thickness overlaps the minimum altitude of the measurement with IWC>0 then the model levels of the cloud are kept constant with the IWC of lowest altitude.*

The vertical information content of the ice parameters measured by CRISTA is limited. If more than one tangent height is affected by clouds in a single profile then the values (IWC and consequently $k_e$ and $\tau$) are used for the interpolation on the vertical grid of SOCRATES. If the predefined cloud thickness $\Delta z$ overlaps the minimum altitude of CRISTA then the model levels of the cloud are kept constant with the IWC of lowest altitude.

- Page 6, lines 155-156: The thicker of the considered clouds overlap with a range of cloud optical thicknesses accessible also by CALIPSO data (at least during night time). This overlap gives an opportunity to test whether the mirroring of August clouds to February makes sense, based on the long and climatologically more relevant CALIPSO measurements.

*This is an excellent idea, but we checked for corresponding publications and did not find seasonal information of CRE for various optical thicknesses. However, Hong et al. (2016) showed seasonal variation of the total SW, LW and net effect of ice clouds over the year and latitudinal resolved (Fig. 4). Although, the cloud optical thickness <0.01 (mainly UTCs) plays only a minor role for the total CALIPSO based CRE it shows only for 60 to 80 ° N significant changes over the seasonal cycle in line with the simplified mirroring of the CRISTA summer measurements to winter conditions. A net warming effect of*

*10–20 W/m² is observed during winter and net cooling effect can exceed 30 W/m² in southern mid to high latitudes and 20 W/m² in the northern mid to high latitudes, with minimum/maximum in August and Jan/Feb. In the tropics, a strong net warming effect (10–20 W/m²) persists over the whole year, mainly caused by high ice clouds (Hong et al., 2016).*

*We added a paragraph based on the text above in the Discussion Section 4.*

- Page 6, lines 157-164: I don't understand fully understand why two methods are used to estimate cloud optical depth.

*Two methods are used, because both method have limited accuracy due to different simplifications in the application process. A comparison of both methods gives confidence about the rough quality of the parameter. A comment has been added in the data preparation section.*

- Page 7, Figure 3: For clarity, I suggest using values instead of log(IWC) or log(COD)

*We changed this to exponential format, see updated Figure 3*

- This was where I first realized that CRISTA dos not allow for an estimation of the vertical extent. Please state that more clearly in the text.

*We added an paragraph in Section 2.3 where we are highlighting this limitation:*

Finally, it needs to be highlighted that the cloud base height is difficult to retrieve for limb measurements. Obviously, for optically thick clouds in the limb (and nadir) direction the cloud bottom is not visible. In addition, for optically thinner clouds limited altitude coverage ($h_{min} >$ CBH) makes it impossible to determine a CBH for many limb sounders (e. g. MIPAS, CRISTA) especially for vertically thick clouds. Consequently, for the sensitivity study with SOCRATES we decided to use predefined cloud thicknesses ($\Delta z$) of 0.5 and 2 km.

- Why are some cloud layers at temperatures above 240 K? Those are likely several km below the tropopause.

*Correct, these events are between 2 to 4 km below the tropopause but at a cloud top height of 8 to 9 km which is not that low for measurements at relatively high latitudes (60-70N). We applied no pre-selection based on temperature for the CRISTA cloud detection, consequently a few (11 out of 162) lower altitude and warm cirrus are included in the calculations. These aspects are now briefly explained in the revised manuscript under data preparation.*

- Page 8, lines 184-185: The mentioned point is one of the main reasons why I think there is only little value in estimating the radiative effects of UTC in February.

  The clouds analyzed seem to end up high above the tropopause altitude if I understand correctly. However, even if one were to correct for this bias, I still think that mirroring the results to February is not much better than guessing.

*We follow the suggestion to move the winter condition results into the discussion section and highlighted the difficulties of our approach very clearly (see also replies further up).*

- Page 11, Fig. 5: Why are just results for aggregates shown for February conditions, and not the more relevant spheric shapes? Mention CRE = 0 line.

*The selection highlights for aggregates the shift from cooling to warming for summer to winter conditions. The change for spherical particles is less pronounced and was therefore not shown. In the revised manuscript we will show now the new hexagonal, aggregates and spherical particle types for the FEB winter results in the new Figure 6. The results will be discussed in Section 4 (like suggested by the reviewer).*

- Page 12, line 231: Why?

*We deleted this sentence ('no correlation') because it is not correct. Already in the original manuscript we highlighted specific region where correlation and linear trends are existing.*

- Page 12, lines 237-238: LW CRE is related to the difference between the surface and cloud top temperatures, not necessarily the cloud temperature itself.

- Page 13, line 238: Lower temperatures at the tropopause -> compared to surface temperatures.

- This is why high clouds in the tropics (to a first approximation) have larger LW CRE than clouds of equivalent optical depth and cloud top temperatures in the extratropics. [consider surface temperatures of 28 °C in the tropics vs. 10 °C in the extratropics].

*We have considered the fact of surface temperature in the analysis by showing now the temperature difference between cloud top and surface temperature in the new Figure 8. Obviously, you can find a correlation with CRE, especially for optically thicker clouds ($\Delta z = 2$ km). For thinner clouds the correlations of spherical particles is significantly less compact.*

- Page 13, line 261: Don't forget the impact of surface albedo! (or the albedo of lower-lying clouds)

*We changed accordingly in Sec.3:*

For winter (February) conditions the short wave cooling becomes smaller due to reduced daylight hours (taking also into account the rather high geographical latitudes) which results in a general warming effect for all winter scenarios and the change from cooling to warming for aggregates and hexagonal particles with $R_{\mathrm{eff}}$ = 10 $\mu$m (Fig. 6 c, d). In addition the mean albedo is changing drastically from summer to winter for the high latitudes of interest from $\alpha$ = 0.15 to 0.5. For details on the portability of the observations to February conditions see also the discussion section 4.

- Page 14, lines 290-292: It may make sense to estimate CRE for 3 cloud thicknesses instead of only 2: the lower plausible thickness (as detected by WISE or other observations), the most frequent one (500 m), and the largest one (2 km).

*We do not think that this is really a necessary extension of the parameter settings. By the additional particle type and the smaller particle size mode the number of settings is already expanded by a factor >2 with strong impact on the results. An additional cloud vertical thickness will in a first glance only influence the optical depth of the selected CRISTA profiles, which are already continuously covering the parameter space of interest (see for example Fig. 6, $\tau$ = 0.03 to 0.0003).*

- Page 15, lines 318-321: Cirrus cases studied are most likely not representative of clouds all over the world, but only of a tiny region in Eurasia. As previously mentioned, I don't think one can easily extrapolate a global effect from that sample of clouds.

*Sensitivity studies with radiation models for cirrus clouds, aerosol, or contrail cirrus are frequently using idealised input profiles for cloud parameter. These parameters are not appropriate all over the world, but by changing the parameters in a realistic range they allow the sensitivity of the radiative effect on the selected parameters to be tested. An estimate of global radiative effect is possible, if for example the global cloud fraction is retrievable by complementary measurements. Here, we applied a slightly different approach by taken selected localised satellite observations (similar to studies using ground stations) for some measurement based input of cloud parameter (in our case cloud top height and IWC/IWP) for the computation of the cloud radiative effect of UTCs. We think it is permitted to do some first estimates on the potential global effect in a similar way like in the literature for contrail cirrus or aerosols (e.g. Myhre et al., 2009b,a). However, we tried to present the discussions in a very distinctive way by using expressions like 'estimate' or 'approximation'. In addition, we have highlighted already in the abstract the restricted region of the input profiles for the SOCRATES sensitivity study on CRE and finally changed the title of the manuscript to highlight the restricted portability of the results.*

- 185. seems

corrected.

[revised manuscript text omitted]

---

## Referee Report (RR1)

**Review of Spang et al, 2023, round 2**

I thank the authors for incorporating many of the suggestions, both clarifications and some major additions or changes (e.g., moving the winter radiation effects to the discussion part of the manuscript). I have two additional major points and several minor points that should be addressed before the study can be published

**1. UTC and climate models**

In the Abstract:
*"These clouds have a small vertical extent and optical depth, and are frequently neither observed even by sensitive sensors nor considered in climate model simulations"*

Such clouds are simulated by at least some models, and are certainly not actively excluded from simulations. Saying "not considered" implies that models just don't simulate them for some reason. This is not true, and should be removed from the sentence.

*"The properties of ultrathin cirrus clouds in the lowermost stratosphere and tropopause region need to be better observed and ultra thin cirrus clouds need to be considered in climate model simulations."*

Instead of considered I suggest "evaluated" or similar.

In conclusions:

*So far UTCs are an unnoticed cirrus cloud type in many fields from cloud microphysical modelling to the parameterisation of the formation processes for various types in global models. …*
*Finally, such work should allow a more accurate quantification of the cooling or warming potential of UTCs*

I would like to point out again that UTCs have not gone unnoticed in climate modeling. They are also not a separate cloud type, just the thinnest of the clouds that form. I believe that such clouds are relatively well sampled in some of the in-situ datasets (indeed, with rather poor coverage over high latitude areas).
Beyond the references mentioned earlier (Gasparini et al., 2018, Nugent et al., 2022, Turbeville et al, 2022, Gasparini et al., 2022), see also Sullivan et al., 2022, Atlas et al., 2023 or Lamraoui et al., 2023, and probably some more that I didn't list.

**2. Other impacts of UTC**

Related to two of the comments in the first round of review:
(1)
*…However, even such thin clouds may be important in modulating the radiative budget within the atmosphere and thus modifying the mean tropopause temperature. In this way, they may indirectly control the amount of water vapor entering the stratosphere (at least in the tropics) and influence circulation patterns. This may be their most important climatic role and should not be neglected in this analysis.*
And (2)
*I would instead appreciate it if the authors showed the heating rates (units K day-1) in the result section. I am mentioning that particularly as it is very straightforward to transform radiative flux vertical profiles in W m-2 to heating rates. High cloud heating rate was shown*

*to strongly modulate circulation patterns (e.g. Voigt et al., 2021, Gasparini et al., 2023 and references therein).*

I suggest adding some information on this issue in the Discussion/Conclusion section as possible follow-up work that could have a larger climate impact.

While the manuscript is very valuable because it is the first to evaluate the CRE of UTC, its main results (for a climate modeler) imply that we shouldn't worry too much about UTC. Their CRE is small, but comparable to the CRE of cirrus contrails. However, in contrast to UTC, contrail cirrus coverage is increasing and contributes significantly to anthropogenic climate forcing.

**Minor points**

1. Line 12:
   I suggest mentioning that you extrapolated the results to winter months from your summer results (instead of "in a more limited way")

2. Introduction: Sherwood et al., 2020 may be an additional good recent reference for cloud feedbacks

3. Line 37-39: The text flow is broken at the sentence starting with "*Luo et al. (2003)…*" and could be fixed.

4. Introduction part related to measurements of thin cirrus: the authors could mention a just published ACPD manuscript that also focuses on cirrus, too thin to be detected by CALIPSO (Lesigne et al., 2023).

5. Line 106: Avery et al., 2012 is a more appropriate reference for CALIOP

6. Figure 4: What exactly are the "ALL" lines representing? I don't see that mentioned in text/caption.

7. The newly written sentence is unclear:

*Where along the limb path (line of sight) the cloud is located, for example in front of or behind the tangent point, and how long the cloud is extended along the line sight is unknown in limb measurements and cannot be retrieved.*

A (shortened, but simplified) suggestion you could consider:

The exact position of the cloud along the limb path (line of sight) remains unknown in limb measurements and is not retrievable.

**References**

Atlas et al., 2023, https://essopenarchive.org/doi/full/10.1002/essoar.10511104.1
Avery et al., 2012, https://doi.org/10.1029/2011GL050545
Lamraoui et al., 2023, https://doi.org/10.5194/acp-23-2393-2023
Lesigne et al., 2023, https://doi.org/10.5194/egusphere-2023-2763
Sherwood et al., 2020, https://doi.org/10.1029/2019RG000678
Sullivan et al., 2022, https://doi.org/10.1029/2022MS003226

---

## Author Response (AR2)

**Radiative impact of thin cirrus clouds in the extratropical lowermost stratosphere and tropopause region egusphere-2023-1234**

Reinhold Spang, Rolf Müller, Alexandru Rap

December 07, 2023

**1 General remark**

We like to thank Blaž Gasparini for reviewing the revised version and for his valuable comments and suggestions which finally improved the manuscript and especially the presentation of the scientific context of the study.

**2 Point-by-point reply to the review of the revised version by Blaž Gasparini:**

- 1. UTC and climate models

  Abstract: *We followed the reviewer suggestion and replaced the term 'considered' with 'evaluated´ on both criticised text passages.*

  Conclusions: *We followed here the suggestion to delete the following passage:* '... which is so far not included in climate models.'

- 2. Other impacts of UTC

  Discussion: *we followed again the suggestion of the reviewer by adding a short paragraph on the potential influence of ultra thin cirrus on circulation patterns in the discussion section under the new item 4.4.*

  4.4 Possible impact on circulation patterns

  Such optically thin clouds, like those reported above, may be important for the radiation budget of the upper troposphere and lower stratosphere and thus for modifying the mean tropopause temperature. They may thus indirectly control the amount of water vapour entering the stratosphere and influence circulation pattern. Radiative vertical flux profiles could be converted to heating rates (e.g.

Kato et al., 2019), and a large cloud heating rate was shown to strongly modulate circulation patterns (e.g. Voigt et al., 2021; Gasparini et al., 2023). This activity could be the topic of follow-up work of the present study.

**Minor points**

- 1. Line 12: I suggest mentioning that you extrapolated the results to winter months from your summer results (instead of "in a more limited way")

  *we changed the sentence:*

  Using sensitivity simulations with different ice effective particle size and shape, we provide an estimate for the uncertainty of the radiative effect of ultra thin cirrus in the extratropical lowermost stratosphere and tropopause region during summer and – by extrapolation of the summer results – for winter.

- 2. Introduction: *We added Sherwood et al. (2020) as suggested.*

- 3. Line 37-39: The text flow is broken at the sentence starting with "Luo et al. (2003)..." and could be fixed.

  *We corrected the sentences like presented here:*

  The dehydration potential of UTTCs in the tropics was shown by Luo et al. (2003). Cirrus clouds in the tropopause region may have a general and significant imprint on the water vapour amount in the stratosphere, and consequently via radiation effects of the stratospheric water vapour on the surface temperature (Riese et al., 2012).

- 4. Introduction part related to measurements of thin cirrus: the authors could mention a just published ACPD manuscript that also focuses on cirrus, too thin to be detected by CALIPSO (Lesigne et al., 2023).

  *We skipped this suggestion, because we don't like to overemphasize in the introduction the difficulties of CALIOP with ultra this cirrus detection.*

- 5. Line 106: Avery et al., 2012 is a more appropriate reference for CALIOP.

  *We replaced the reference.*

- 6. Figure 4: What exactly are the "ALL" lines representing? I don't see that mentioned in text/caption.

  *We explained this now in more detail in the caption of Figure 4:*

  ... separation in cloud thickness $\Delta z = 0.5$ km and $\Delta z = 2$ km in blue and red for spherical/hexagonal/spherical (a-c) (the total of $\Delta z = 0.5$ and $2.0$ km is highlighted in black and labeled with ALL) ...

- 7. The newly written sentence is unclear: A (shortened, but simplified) suggestion you could consider: 'The exact position of the cloud along the limb path (line of sight) remains unknown in limb measurements and is not retrievable.'

  *We replaced the sentence with exactly this suggestion.*

**References**

Gasparini, B., Sullivan, S. C., Sokol, A. B., Kärcher, B., Jensen, E., and Hartmann, D. L.: Opinion: Tropical cirrus — From micro-scale processes to climate-scale impacts, EGUsphere, 2023, 1–47, https://doi.org/10.5194/egusphere-2023-1214, 2023.

Kato, S., Rose, F. G., Ham, S. H., Rutan, D. A., Radkevich, A., Caldwell, T. E., Sun-Mack, S., Miller, W. F., and Chen, Y.: Radiative Heating Rates Computed With Clouds Derived From Satellite-Based Passive and Active Sensors and their Effects on Generation of Available Potential Energy, Journal of Geophysical Research: Atmospheres, 124, 1720–1740, https://doi.org/https://doi.org/10.1029/2018JD028878, 2019.

Luo, B. P., Peter, T., Fueglistaler, S., Wernli, H., Wirth, M., Kiemle, C., Flentje, H., Yushkov, V. A., Khattatov, V., Rudakov, V., Thomas, A., Borrmann, S., Toci, G., Mazzinghi, P., Beuermann, J., Schiller, C., Cairo, F., Di Donfrancesco, G., Adriani, A., Volk, C. M., Strom, J., Noone, K., Mitev, V., MacKenzie, R. A., Carslaw, K. S., Trautmann, T., Santacesaria, V., and Stefanutti, L.: Dehydration potential of ultrathin clouds at the tropical tropopause, Geophysical Research Letters, 30, https://doi.org/https://doi.org/10.1029/2002GL016737, 2003.

Riese, M., Ploeger, F., Rap, A., Vogel, B., Konopka, P, Dameris, M., and Forster, P.: Impact of uncertainties in atmospheric mixing on simulated UTLS composition and related radiative effects, J. Geophys. Res., 117, D16305, https://doi.org/10.1029/2012JD017751, 2012.

Voigt, A., Albern, N., Ceppi, P., Grise, K., Li, Y., and Medeiros, B.: Clouds, radiation, and atmospheric circulation in the present-day climate and under climate change, WIREs Climate Change, 12, e694, https://doi.org/https://doi.org/10.1002/wcc.694, 2021.